# PS-PPO: Prefix-Sampling PPO for Critic-Free RLHF

**Doo Hwan Hwang** [1]  **Kee-Eung Kim** [1]

## Abstract

Reinforcement Learning from Human Feedback (RLHF) for Large Language Models increasingly relies on critic-free methods as a practical alternative to actor–critic training. Despite their simplicity, existing critic-free approaches propagate a trajectory-level learning signal uniformly across all tokens in a trajectory. This requires full-trajectory policy updates for every rollout, leading to substantial optimization cost for long reasoning traces, even though intermediate prefixes often contain enough information to largely determine the final outcome. We propose Prefix-Sampling Proximal Policy Optimization (PS-PPO), a compute-efficient critic-free method for RLHF that exploits this temporal redundancy. PS-PPO introduces a prompt-conditioned cutoff distribution and samples a cutoff timestep for each trajectory. During the update pass, PS-PPO backpropagates only through the sampled prefix of each trajectory and applies an importance-weighting correction so that the resulting truncated gradient estimator remains unbiased with respect to the full-trajectory objective. Experiments on mathematical reasoning and RLHF benchmarks show that PS-PPO achieves large reductions in training compute and peak GPU memory, while maintaining accuracy comparable to strong critic-free baselines.

## 1. Introduction

As Large Language Models (LLMs) are increasingly deployed for complex reasoning and decision-making tasks, the quality of a generated response is determined not merely by linguistic fluency, but by its alignment with human preferences and task objectives (Radford et al., 2019; Brown et al., 2020; Anil et al., 2023; Wang et al., 2024; Jaech

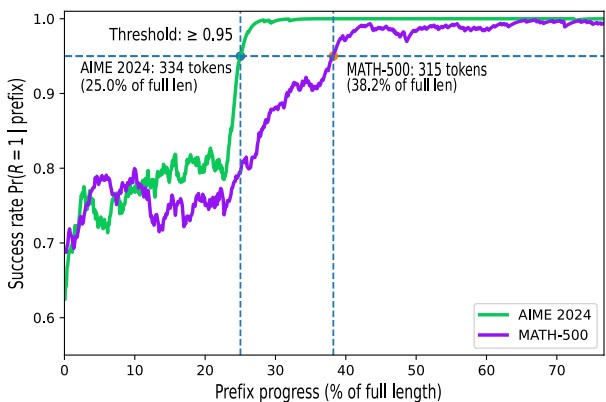

*Figure 1.* Prefix-conditioned success rate on AIME 2024 and MATH-500 versus prefix progress, where prefix progress denotes the percentage of the full completion length. We estimate the success rate using Qwen2.5-Math-7B with 32 suffix rollouts per prefix. The success rate often stabilizes well before the end of the full completion.

et al., 2024). Since such objectives are difficult to encode directly in a token-level likelihood objective, Reinforcement Learning from Human Feedback (RLHF) has become a standard method for aligning model behavior with desired outcomes (Ouyang et al., 2022; Wang et al., 2024). In practice, RLHF typically assigns a single scalar reward after a full completion is generated. This delayed feedback provides little information about which parts of a long trajectory were responsible for success or failure, making temporal credit assignment a central challenge (Arjona-Medina et al., 2019; Harutyunyan et al., 2019; Li et al., 2025).

A classic remedy is to introduce a critic network to estimate token-level value functions and provide the value estimates to support policy-gradient optimization, such as Proximal Policy Optimization (PPO) (Schulman et al., 2017). However, training a critic at LLM scale incurs substantial compute and memory overhead and is often a major source of optimization instability (Shao et al., 2024; Hao et al., 2025; Kazemnejad et al., 2025). These limitations have motivated critic-free alternatives such as GRPO (Shao et al., 2024) and RLOO (Ahmadian et al., 2024), which avoid explicit value learning by estimating advantages from multiple trajectories sampled per prompt. However, they commonly broadcast the reward signal uniformly across all timesteps (Lambert,

[1]Kim Jaechul Graduate School of AI, KAIST, Daejeon, South Korea. Correspondence to: Doo Hwan Hwang <dhhwang@ai.kaist.ac.kr>.

*Proceedings of the 43rd International Conference on Machine Learning*, Seoul, South Korea. PMLR 306, 2026. Copyright 2026 by the author(s).

2024), requiring full-trajectory updates that lead to substantial compute cost for long reasoning traces.

Our motivation for addressing this full-trajectory bottleneck comes from the observation that, in many reasoning tasks, intermediate prefixes already contain substantial information about the final outcome. For example, in step-by-step mathematical reasoning, early decisions such as the choice of decomposition or intermediate derivation often strongly constrain whether the final answer can be correct. Later tokens may still be useful, especially when a model revises or corrects its earlier reasoning. However, our late-recovery analysis in Appendix A suggests that such cases are not the dominant pattern in the reasoning traces we study. We empirically support this observation by analyzing success rates conditioned on intermediate prefixes, as shown in Figure 1. Across tasks, prefix-conditioned success rates often saturate before the end of the completion, indicating that intermediate prefixes can be highly predictive of the final outcome. Building on this observation, we propose **Prefix-Sampling Proximal Policy Optimization (PS-PPO)**, a compute-efficient critic-free method. PS-PPO defines a prompt-dependent cutoff distribution of timesteps from a prefix uncertainty signal, retaining only the prefix up to the sampled cutoff during the policy gradient update. To correct the randomness introduced by truncation, we derive a reweighted estimator whose expectation over cutoff sampling recovers the corresponding full critic-free broadcast update.

Our main contributions are as follows:

- We propose PS-PPO, a compute-efficient critic-free method that samples a prompt-dependent cutoff and computationally truncates forward *and* backward computation beyond the cutoff.

- We derive an inclusion-probability reweighted estimator that corrects for stochastic truncation and recovers the full broadcast update in expectation over cutoff sampling, without additional rollouts or auxiliary models.

- We demonstrate that PS-PPO reduces training-time compute and peak GPU memory while maintaining accuracy relative to strong critic-free baselines on mathematical reasoning and RLHF benchmarks.

## 2. Related Work

### 2.1. Post-Training for LLM Alignment

Post-training of pretrained LLMs is typically conducted via Supervised Fine-Tuning (SFT) or Reinforcement Learning (RL) (Grattafiori et al., 2024; Trung et al., 2024). While SFT relies on imitating static datasets provided by humans,

RL offers a distinct advantage by optimizing for reward maximization. This approach reduces the dependency on extensive ground-truth demonstrations and encourages the model to explore diverse solution paths beyond the training data distribution (Shao et al., 2024; Jaech et al., 2024). This capability has established a new paradigm for reasoning models, enabling them to solve complex problems through broad exploration.

### 2.2. Critic-Free Policy Optimization in LLMs

Standard RLHF pipelines commonly employ actor–critic algorithms to optimize a policy with respect to scalar rewards produced by a reward model. While effective, scaling actor–critic training to large language models often requires substantial memory and compute, and the resulting updates can be sensitive to value estimation accuracy, potentially affecting training stability (Gao et al., 2024).

Motivated by these challenges, critic-free alternatives such as RLOO (Ahmadian et al., 2024), GRPO (Shao et al., 2024), Dr.GRPO (Liu et al., 2025), and DAPO (Yu et al., 2025) have been proposed. In these methods, the parameterized value network is removed and advantages are computed from multiple samples generated for the same prompt. They broadcast a single trajectory-level learning signal across all tokens, which requires backpropagating through the entire completion. This makes the update increasingly expensive as completions grow longer, motivating token-selective policy optimization.

### 2.3. Token-Selective RL for LLMs

To mitigate the inefficiency of full-sequence updates, recent work has explored token-selective policy optimization by applying learning signals only to a selected subset of tokens. Wang et al. (2025) identify a small fraction of high-entropy tokens in chain-of-thought trajectories as decision points and restrict policy updates to those positions. S-GRPO (Lee & Tong, 2025) keeps a heuristically chosen fixed prefix and subsamples the remaining suffix tokens using Bernoulli sampling. PPPO (Sun et al., 2025) updates only prefix tokens by progressively increasing the prefix length according to a predefined schedule during training, and evaluates each prefix by sampling multiple continuations conditioned on that prefix and aggregating their rewards. These methods are commonly implemented via gradient masking, which can yield limited computational savings when forward and backward passes are still executed over the full sequence. In contrast, PS-PPO avoids fixed or heuristic token-selection rules by deriving a prompt-dependent cutoff distribution in a principled way. It then samples a cutoff position from this distribution and backpropagates only through the sampled prefix, directly reducing backpropagation cost.

## 3. Method

### 3.1. Background

Policy optimization (Ouyang et al., 2022) is a primary framework for the post-training of language model policies. The objective is to generate a completion $o$ that maximizes the expected reward $R(x, o)$ given a prompt $x$ from a query distribution $p_Q$. Formally, let $o = [o_1, \ldots, o_T]$ represent a sequence of length $T$, where each token $o_t$ belongs to the vocabulary $\mathcal{V}$. We define the prefix state at timestep $t$ as $s_t = [x, o_1, \ldots, o_{t-1}]$ and the suffix tokens after timestep $t$ as $o_{>t} = [o_{t+1}, \ldots, o_T]$.

To optimize the policy $\pi_\theta$, we adopt the standard PPO training objective:

$$
\mathcal{J}_{\text{PPO}}(\theta) = \mathbb{E}_{x \sim p_Q, o \sim \pi_{\theta_{\text{old}}}(\cdot | x)}
$$
$$
\left[ \sum_{t=1}^{T} \min \left( \rho_t(\theta) \, \hat{A}_t, \, \text{clip}(\rho_t(\theta), 1 - \epsilon, 1 + \epsilon) \hat{A}_t \right) \right] \quad (1)
$$

where $\rho_t(\theta) = \frac{\pi_\theta(o_t | s_t)}{\pi_{\theta_{\text{old}}}(o_t | s_t)}$ is the per-timestep importance weight, $\hat{A}_t$ is the estimated advantage at timestep $t$, and $\epsilon$ is the clipping range. The advantage $A_t^\pi$ quantifies the relative benefit of selecting token $o_t$ compared to the expected reward at the current state $s_t$. It is formally defined as:

$$
A_t^\pi = Q_t^\pi - V_t^\pi,
$$

where $Q_t^\pi := \mathbb{E}_\pi[R \mid s_t, o_t]$ is the action-value function and $V_t^\pi := \mathbb{E}_\pi[R \mid s_t]$ is the state-value function. In the context of RLHF, rather than using per-token value estimates, it is common to use a broadcast advantage estimator:

$$
\hat{A}_t = R(x, o) - b(x),
$$

which is shared across all timesteps within a completion (Shao et al., 2024). Here, the baseline $b(x)$ is typically estimated as the mean reward over multiple completions sampled for the same prompt $x$.

While the broadcast advantage estimator assigns the same scalar weight $\hat{A}_t$ to every token in the completion $o$, this does not imply that all timesteps are equally informative about the final outcome. As illustrated in Figure 1, many trajectories become highly predictive before the completion ends, suggesting that full-sequence updates in Eq. (1) can spend substantial computation and memory on suffix tokens with redundant marginal learning signal. Importantly, later tokens may still matter in self-correcting traces; PS-PPO therefore uses stochastic truncation rather than deterministically discarding all suffix tokens.

To enhance computational efficiency, we propose a completion truncation method based on prefix sampling. However, applying a fixed truncation length across all completions introduces significant bias; a fixed timestep $t$ does not ensure

that the advantage of the discarded suffix $o_{>t}$ is truly near zero for every prompt-completion pair.

### 3.2. Unbiased Stochastic Truncation

Cutting off completions and performing gradient updates only for truncated prefixes introduce bias. In this section, we introduce a reweighting scheme that ensures the expected update for prefixes matches that of a full-sequence update.

We begin by assuming that the minimum prefix length $H$ required for policy optimization is a random variable governed by a survival function, which is in turn determined by a non-increasing sequence of cutoff probabilities.

$$
1 \geq \xi_1 \geq \xi_2 \geq \cdots \geq \xi_T > 0.
$$

The quantity $\xi_t$ denotes the probability that the $t$-th token $o_t$ is retained during training, or the probability that the effective truncation point $H$ is at least $t$ (i.e., $\Pr(H \geq t \mid x) = \xi_t$). Under this framework, we backpropagate gradients only through the prefix tokens $o_t$ where $t \leq H$, $\mathbb{E}_{x, o \sim \pi_\theta, H}[\sum_{t=1}^{H} \hat{A}_t \nabla_\theta \log \pi_\theta(o_t | s_t)]$. However, simply truncating the summation at $H$ results in a biased estimator of the full update. To construct an unbiased estimator, we must reweight the gradients. First, let us define the per-timestep policy gradient for completion $o^{(k)}$ at timestep $t$ over the group $K$ assuming the use of the GRPO method:

$$
g_t^{(k)}(\theta) := \hat{A}_t^{(k)} \nabla_\theta \log \pi_\theta\left(o_t^{(k)} \mid s_t^{(k)}\right).
$$

The full update direction for prompt $x$ is then

$$
G(\theta) := \frac{1}{K} \sum_{k=1}^{K} \sum_{t=1}^{T} g_t^{(k)}(\theta). \quad (2)
$$

Given sampled cutoffs $\{H^{(k)}\}_{k=1}^{K}$, we form the truncated yet unbiased estimator via reweighting the gradients,

$$
\widehat{G}(\theta) := \frac{1}{K} \sum_{k=1}^{K} \sum_{t=1}^{H^{(k)}} \frac{1}{\xi_t} g_t^{(k)}(\theta). \quad (3)
$$

However, this reweighting introduces extra variance (see Appendix B for the derivation).

### 3.3. Budgeted Monotone Cutoff Distribution

Our goal is to choose a cutoff distribution $\xi_{1:T}$ that minimizes the additional update variance induced by the importance sampling weights, subject to a gradient update compute budget $B$. The optimization problem is formulated

as:

$$\min_{\xi_{1:T}} \mathbb{E}\left[\operatorname{tr}\!\left(\operatorname{Cov}_H\!\left(\widehat{G}(\theta)\mid x\right)\right)\right]$$

$$\text{s.t.} \quad \sum_{t=1}^{T} t\left(\xi_t - \xi_{t+1}\right) = B, \quad \xi_{T+1} := 0,$$

$$1 \geq \xi_1 \geq \cdots \geq \xi_T > 0.$$

The budget $B$ constrains the expected length of prefixes for which we perform gradient updates, $\sum_{t=1}^{T} t\left(\xi_t - \xi_{t+1}\right)$, where $\xi_{T+1} := 0$ for notational simplicity.

For tractability, we use a timestep-wise trace surrogate of the exact cutoff-induced variance. The exact expansion contains cross-timestep covariance terms due to autoregressive dependence, but including them would couple the cutoff probabilities across timesteps and obscure a timestep-wise allocation of the update budget. We therefore retain the diagonal trace terms as a tractable surrogate, yielding the following approximation (derivation in Appendix C):

$$\mathbb{E}\left[\operatorname{tr}\!\left(\operatorname{Cov}_H\!\left(\widehat{G}(\theta)\mid x\right)\right)\right] \approx \frac{1}{K}\sum_{t=1}^{T} w_t^\theta(x)\left(\frac{1}{\xi_t}-1\right). \tag{4}$$

where $w_t^\theta(x)$ is defined by

$$w_t^\theta(x) := \mathbb{E}\left[\left\|g_t(\theta)\right\|^2 \,\middle|\, x\right]$$
$$= \mathbb{E}\left[\hat{A}_t^2 \left\|\nabla_\theta \log \pi_\theta(o_t \mid s_t)\right\|^2 \,\middle|\, x\right].$$

Minimizing (4) with respect to $\xi_{1:T}$ is equivalent to solving

$$\min_{\xi_{1:T}} \sum_{t=1}^{T} \frac{w_t^\theta(x)}{\xi_t}. \tag{5}$$

To compute Eq. (5), we must estimate the weights $w_t^\theta(x)$. However, $w_t^\theta(x)$ involves the score-norm factor $\left\|\nabla_\theta \log \pi_\theta(o_t \mid s_t)\right\|^2$, and computing it exactly would require full backward passes, offsetting the compute savings from truncation. We therefore construct a forward-only proxy from the model's output head.

**Forward-only proxy of $w_t^\theta(x)$.** The score gradient $\nabla_\theta \log \pi_\theta(o_t \mid s_t)$ requires a full backpropagation pass through the entire network. However, for computing the cutoff probabilities, we do not need the exact score norm at every timestep. Rather, we need a tractable signal that captures the relative update importance of different timesteps. We therefore use the gradient at the output head, $\nabla_{\theta_{\text{out}}} \log \pi_\theta(o_t \mid s_t)$, as a forward-computable proxy for the score-norm factor in $w_t^\theta(x)$:

$$\left\|\nabla_\theta \log \pi_\theta(o_t \mid s_t)\right\|^2 \approx \left\|\nabla_{\theta_{\text{out}}} \log \pi_\theta(o_t \mid s_t)\right\|^2.$$

This approximation is motivated by our empirical finding that the output-head score norm is strongly correlated with the full-parameter score norm across timesteps; the correlation analysis is reported in Appendix H.

For a standard softmax head in the last layer, the norm $\left\|\nabla_{\theta_{\text{out}}} \log \pi_\theta(o_t \mid s_t)\right\|^2$ has the following closed-form expression using only forward-pass activations:

$$\gamma_t(s_t, o_t) := \|h_t\|_2^2 \left(1 - 2\pi_{\theta_{\text{old}}}(o_t \mid s_t) + \|\pi_{\theta_{\text{old}}}(\cdot \mid s_t)\|_2^2\right). \tag{6}$$

See Appendix D for the derivation. We then define the proxy-based weights as

$$\tilde{w}_t(x) := \mathbb{E}\left[\hat{A}_t^2 \, \gamma_t(s_t, o_t) \,\middle|\, x\right].$$

Both approximations are used only to compute the cutoff probabilities $\xi_{1:T}$. They do not affect the unbiasedness of the truncated estimator itself.

In critic-free methods, the advantage is broadcast within a completion, so $\hat{A}_t = \hat{A} = R - b(x)$ for all $t$ in that completion. Using the law of total expectation over the prefix state $s_t$ and a mean-field approximation, we obtain

$$\tilde{w}_t(x) := \mathbb{E}\left[(R - b(x))^2 \, \gamma_t(s_t, o_t) \,\middle|\, x\right]$$
$$\approx \mathbb{E}_{s_t|x}\left[\mathbb{E}\left[(R - b(x))^2 \,\middle|\, s_t\right]\right] \cdot \underbrace{\mathbb{E}_{s_t|x, o_t \sim \pi_{\theta_{\text{old}}}(\cdot|s_t)}\left[\gamma_t(s_t, o_t)\right]}_{=:\, \bar{\gamma}_t(x,t)}. \tag{7}$$

To estimate $E_{s_t|x}[(R - b(x))^2 | s_t]$, we consider the upper bound of reward variance $\operatorname{Var}(R \mid s_t)$. This is because the baseline $b(x)$ is typically close to the expected reward $\mathbb{E}[R|s_t]$, resulting in near-zero bias. For binary rewards, we use the following reward-uncertainty proxy:

$$u_t(x) := \frac{1}{2} - \frac{1}{2}\left\|p(x)\bar{\pi}_G(\cdot \mid t, x) - (1 - p(x))\bar{\pi}_B(\cdot \mid t, x)\right\|_1,$$

where $p(x) = \Pr(R = 1 \mid x)$, and $\bar{\pi}_G$ and $\bar{\pi}_B$ denote the state-averaged next-token distributions over successful and unsuccessful rollouts, respectively. Appendix E shows that $\mathbb{E}[\operatorname{Var}(R \mid S_t) \mid x] \leq u_t(x)$. Therefore, we use the reward uncertainty at each timestep $t$ for the advantage when optimizing the cutoff distribution $\xi_{1:T}$.

**Final optimization problem.** Consequently, we solve the following convex optimization problem:

$$\arg\min_{\xi_{1:T}} \sum_{t=1}^{T} \frac{\bar{\gamma}_t(x,t)u_t(x)}{\xi_t} \tag{8}$$

$$\text{s.t.} \quad \sum_{t=1}^{T} t\left(\xi_t - \xi_{t+1}\right) = B, \quad \xi_{T+1} := 0,$$

$$1 \geq \xi_1 \geq \cdots \geq \xi_T > 0.$$

The uncertainty term $u_t(x)$ measures the residual uncertainty of the terminal reward after observing the prefix state $s_t$. When $u_t(x)$ is small, the prefix already makes the final reward largely predictable, so the corresponding timestep can be sampled less frequently. Conversely, large $u_t(x)$ indicates that the terminal outcome remains uncertain, so skipping that timestep would induce larger variance. The design objective therefore assigns larger inclusion probabilities to timesteps with larger $\bar{\gamma}_t(x,t)u_t(x)$, while allowing predictable regions to be truncated more aggressively under the compute budget. This yields shorter sampled prefixes once the reward becomes sufficiently predictable, without changing the full-sequence update in expectation.

Directly optimizing Eq. (8) is challenging due to the monotonicity constraint. To address this within this convex problem, we employ the Pooling Adjacent Violations (PAV) algorithm (Brummer & du Preez, 2013), which returns the optimal monotone solution. We begin by solving the Lagrangian relaxation of the problem Eq. (8) without the monotonicity constraint, yielding the closed-form solution:

$$\xi_t = B \cdot \frac{\sqrt{\bar{\gamma}_t(x,t)u_t(x)}}{\sum_{j=1}^{T}\sqrt{\bar{\gamma}_j(x,j)u_j(x)}}.$$

We then apply the PAV algorithm to these initial values to obtain the final monotone optimum $\xi_{1:T}^{\star}$. The complete derivation and the details of the pooling procedure are provided in Appendix F.

Using the optimized $\xi_{1:T}^{\star}$, we independently sample a cutoff $H^{(k)}$ for each completion $k \in \{1, \ldots, K\}$. We then construct an importance-weighted truncated update by reweighting each retained timestep by $1/\xi_t^{\star}$:

$$\widehat{G}^{\text{PS-PPO}}(\theta) := \frac{1}{K}\sum_{k=1}^{K}\sum_{t=1}^{H^{(k)}}\frac{1}{\xi_t^{\star}}g_t^{(k)}(\theta). \qquad (9)$$

The full procedure is summarized in Algorithm 1.

## 4. Experiments

We evaluate our method on a diverse set of math reasoning benchmarks. Our goal is to assess whether the proposed approach yields consistent gains over strong critic-free baselines in terms of task performance and efficiency. We further conduct a sensitivity analysis over key hyperparameters to characterize robustness.

**Experiment setup.** For training on mathematical reasoning, we train on MATH (Hendrycks et al., 2021), excluding Level 1 and Level 2 problems, and use Llama-3.1-

---

**Algorithm 1** Prefix-Sampling PPO (PS-PPO)

**Require**: policy model $\pi_\theta$; prompt distribution $p_Q$; reward function $R(\cdot,\cdot)$; max training steps $N$; old-policy refresh period $F$; group size $K$; max completion length $T$; budget $B$.

1: Initialize policy parameters $\theta$, and set $\theta_{\text{old}} \leftarrow \theta$.
2: **for** iteration $n = 1$ **to** $N$ **do**
3:     Sample a prompt $x \sim p_Q$.
4:     Sample $K$ completions $\{\tau^{(i)}\}_{i=1}^{K}$ from $\pi_{\theta_{\text{old}}}(\cdot \mid x)$.
5:     Compute terminal rewards $\{R^{(i)}\}_{i=1}^{K}$.
6:     Compute critic-free advantages $\{\hat{A}^{(i)}\}_{i=1}^{K}$ using the within-prompt baseline.
7:     Compute per-timestep proxies $\{u_t(x), \bar{\gamma}_t(x,t)\}_{t=1}^{T}$ from $\{(\tau^{(i)}, R^{(i)})\}_{i=1}^{K}$.
8:     Set cutoff weights $w_t(x) \leftarrow \bar{\gamma}_t(x,t)u_t(x)$ for $t = 1, \ldots, T$.
9:     Compute monotone cutoff probabilities $\xi_{1:T}$ by solving the budgeted design problem with constraints.
10:    Sample cutoffs $\{H^{(i)}\}_{i=1}^{K}$ from the cutoff distribution induced by $\xi_{1:T}$.
11:    During the update pass, backpropagate only through tokens $t \leq H^{(i)}$ and reweight token losses by $1/\xi_t$.
12:    Update $\theta$ using PPO with the truncated, reweighted gradient estimator in Eq. (9).
13:    **if** $n \mod F = 0$ **then**
14:       $\theta_{\text{old}} \leftarrow \theta$.
15:    **end if**
16: **end for**
17: **return** optimized policy model $\pi_\theta$.

---

8B-Instruct[1] and Qwen2.5-Math-7B[2] (Yang et al., 2024) as backbone models. Prompts are designed to elicit step-by-step derivations via an explicit instruction, and we optimize the policy with a binary reward based on final-answer correctness. We also evaluate performance on challenging benchmarks: MATH500 (Hendrycks et al., 2021), AMC 2023 (Li et al., 2024), Minerva Math (Lewkowycz et al., 2022), CollegeMath (Tang et al., 2024), and AIME 2024/2025 (Art of Problem Solving, 2026).

Although our main experiments focus on mathematical reasoning, PS-PPO is not tied to binary correctness rewards. The stochastic truncation estimator preserves the corresponding full-sequence update in expectation, so the same update principle applies to RLHF settings with general terminal rewards. We therefore include continuous-reward text-generation experiments in Appendix G, where PS-PPO reduces training time and memory while maintaining competitive reward performance.

---

[1] https://huggingface.co/meta-llama/Llama-3.1-8B-Instruct
[2] https://huggingface.co/Qwen/Qwen2.5-Math-7B

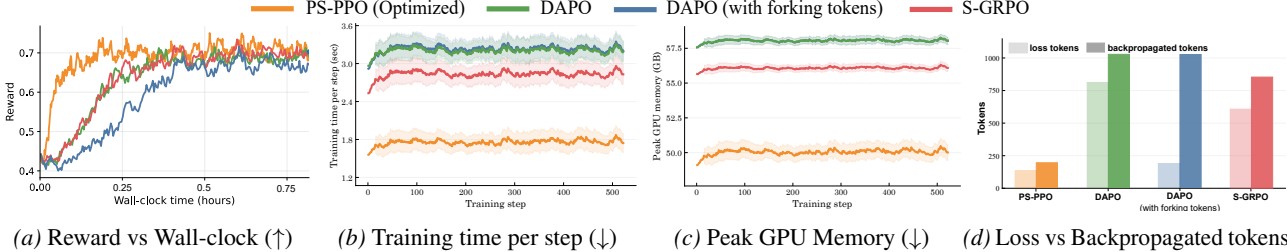

*(a)* Reward vs Wall-clock (↑)     *(b)* Training time per step (↓)     *(c)* Peak GPU Memory (↓)     *(d)* Loss vs Backpropagated tokens

*Figure 2.* Efficiency of PS-PPO: PS-PPO matches baseline performance while reducing training-time compute. We report reward versus wall-clock time, training time per step, peak GPU memory, and loss-applied versus backpropagated tokens. Here, training time denotes the time spent on the gradient-update stage (including computing $\xi_{1:T}$ and the forward/backward passes), excluding rollout/generation. All methods use the same number of training steps with $K=8$ rollouts per prompt and are trained for 3 epochs.

*Table 1.* Training-time breakdown per training step excluding rollout/generation. Values are mean±std over training steps. PS-PPO substantially reduces training time per step primarily by lowering the forward/backward cost, even after accounting for the overhead of computing $\xi_{1:T}(x)$.

| Method | Training-time breakdown (s) | | | | Total training time (s) |
|---|---|---|---|---|---|
| | Computing $\xi_{1:T}$ | Forward | Backward | Other | |
| PS-PPO (Optimized, B=128) | $0.43 \pm 0.02$ | $0.32 \pm 0.04$ | $1.01 \pm 0.14$ | $0.01 \pm 0.01$ | $1.77 \pm 0.02$ |
| S-GRPO | N/A | $0.82 \pm 0.07$ | $1.81 \pm 0.10$ | $0.03 \pm 0.02$ | $2.66 \pm 0.01$ |
| DAPO | N/A | $1.11 \pm 0.04$ | $2.10 \pm 0.13$ | $0.02 \pm 0.01$ | $3.23 \pm 0.01$ |
| DAPO (with forking tokens) | N/A | $1.12 \pm 0.02$ | $2.12 \pm 0.11$ | $0.01 \pm 0.01$ | $3.25 \pm 0.01$ |

**Baselines.** We compare PS-PPO against representative critic-free RLHF baselines and methods that aim to reduce the cost of policy updates. We include GRPO (Shao et al., 2024), Dr.GRPO (Liu et al., 2025), DAPO (Yu et al., 2025), and RLOO (Ahmadian et al., 2024), which backpropagate through the full generated completion.

We also compare to a fixed-prefix baseline that always backpropagates through a predetermined prefix length $L$, as well as update-efficient baselines including DAPO with forking tokens (Wang et al., 2025) and S-GRPO (Lee & Tong, 2025).

To isolate the effect of the cutoff distribution, we evaluate PS-PPO variants that differ only in the cutoff-sampling distribution : (i) **PS-PPO (Uniform)**, which samples the cutoff uniformly over timesteps; (ii) **PS-PPO (Time Prior)**, which sets $\xi_t = \exp(-\lambda(t-1))$ with $\lambda \geq 0$ controlling how strongly the cutoff favors earlier timesteps; (iii) **PS-PPO (Heuristic)**, which first computes an unconstrained sequence of cutoff probabilities $\xi_{1:T}$, and then enforces the required prefix monotonicity via a simple post-processing: $\xi_1 = \xi_1$ and $\xi_t \leftarrow \min(\xi_{t-1}, \xi_t)$ for $t = 2, \ldots, T$, without optimizing the monotone cutoff objective; (iv) **PS-PPO (Optimized)**, which optimizes the cutoff distribution under the prefix monotonicity and compute-budget constraints.

### 4.1. Mathematical Reasoning

**Compute efficiency and performance.** We evaluate PS-PPO on mathematical reasoning benchmarks, asking whether it can reduce training time (excluding roll-out/generation, which is shared across methods) while matching strong critic-free baselines. We use a backpropagation budget of $B=128$, where $B$ denotes the expected number of prefix tokens included in the gradient update. This setting yielded the best accuracy–training-time trade-off in our sweep, and ablations over $B$ are reported in Sec. 4.2. All methods use the same rollout (group size $K$) and the same number of training steps.

Table 2 shows that PS-PPO (Optimized) achieves accuracy comparable to strong critic-free baselines. Importantly, this competitive performance is attained alongside improved computational efficiency: as shown in Fig. 2 and Table 1, PS-PPO reaches high rewards faster while reducing training time and peak GPU memory usage. The savings primarily come from lower forward/backward cost. Although PS-PPO incurs additional overhead to compute the cutoff probabilities $\xi_{1:T}(x)$ for truncation, this cost is outweighed by the savings from shorter backpropagated prefixes, yielding a 33%–45% reduction in training time per step relative to the baselines. Peak GPU memory also decreases because truncation reduces stored activations, reducing peak memory by 15%–17% relative to the non-truncated baselines.

In contrast, masking-only baselines (S-GRPO; DAPO with forking tokens) reduce the number of loss-included tokens (i.e., tokens with non-zero loss after masking; Fig. 2(d)), but do not necessarily reduce the dominant forward/backward cost because computation is still performed over the full sequence length. Accordingly, these baselines yield only modest reductions in training time per step and peak mem-

*Table 2.* Pass@1 accuracy (%) on mathematical reasoning benchmarks. All methods use the same number of training steps and the same number of rollouts per prompt ($K$). Under this fixed rollout, PS-PPO attains competitive accuracy relative to strong critic-free baselines with improved training-time efficiency (Fig. 2, Table 1). Values are averaged over three independent runs. (**Bold**: best, underlined: second best).

| BASE MODEL | METHOD | MATH500 | AMC23 | COLLEGEMATH | MINERVAMATH | AIME24 | AIME25 |
|---|---|---|---|---|---|---|---|
| LLAMA-3.1 8B-INSTRUCT | BASE | 43.2 | 25.0 | 30.1 | 25.7 | 3.3 | 0.0 |
| | GRPO | 45.2 | 30.0 | 32.1 | 27.9 | 6.6 | 0.0 |
| | DR.GRPO | 46.0 | 27.5 | 32.8 | 28.3 | 6.6 | **3.3** |
| | RLOO | 45.2 | 27.5 | 32.1 | 26.1 | 6.6 | 0.0 |
| | DAPO | 46.8 | **32.5** | **34.2** | **29.4** | **10.0** | 3.3 |
| | DAPO (WITH FORKING TOKENS) | 47.0 | **32.5** | 33.4 | 29.2 | **10.0** | 3.3 |
| | S-GRPO | 46.2 | 30.8 | 33.0 | 28.6 | **10.0** | 3.3 |
| | FIXED LENGTH ($L$=512) | 45.5 | 27.5 | 32.2 | 26.7 | 6.6 | 0.0 |
| | PS-PPO (UNIFORM) | 45.8 | 27.8 | 32.6 | 27.0 | 6.6 | 1.1 |
| | PS-PPO (TIME-PRIOR) | 45.6 | 27.6 | 32.5 | 26.9 | 6.6 | 1.1 |
| | PS-PPO (HEURISTIC) | 45.9 | 28.0 | 32.7 | 27.1 | 6.6 | 1.1 |
| | PS-PPO (OPTIMIZED, $B$=128) | **47.6** | **32.5** | 33.5 | 29.2 | **10.0** | 3.3 |
| QWEN-2.5 MATH-7B | BASE | 82.5 | 52.5 | 21.5 | 27.6 | 6.6 | 6.7 |
| | GRPO | 83.9 | 55.8 | 23.7 | 29.4 | 13.3 | 10.0 |
| | DR.GRPO | 83.5 | 57.5 | 23.6 | 29.8 | 13.3 | 10.0 |
| | RLOO | 83.6 | 55.0 | 22.7 | 28.7 | 10.0 | 6.7 |
| | DAPO | **85.2** | 57.5 | 24.2 | **30.5** | **16.7** | 10.0 |
| | DAPO (WITH FORKING TOKENS) | 85.1 | 58.3 | **25.0** | 30.1 | **16.7** | 10.0 |
| | S-GRPO | 84.7 | 56.7 | 22.8 | 29.0 | **16.7** | 10.0 |
| | FIXED LENGTH ($L$=512) | 83.3 | 55.0 | 23.4 | 28.3 | 10.0 | 8.9 |
| | PS-PPO (UNIFORM) | 83.6 | 55.8 | 23.7 | 28.7 | 13.3 | 9.0 |
| | PS-PPO (TIME-PRIOR) | 84.6 | 55.6 | 23.7 | 27.8 | 10.0 | 6.7 |
| | PS-PPO (HEURISTIC) | 83.7 | 56.0 | 22.7 | 27.9 | 13.3 | 10.0 |
| | PS-PPO (OPTIMIZED, $B$=128) | 85.0 | **58.5** | 23.9 | 30.1 | **16.7** | **13.3** |

*Table 3.* Scaling with maximum completion length. Training time per step excludes rollout/generation and measures the update stage. Accuracy is pass@1. Avg. is averaged over MATH500, AIME24, and AIME25.

| $T_{max}$ | Method | Time/step | MATH500 | AIME24 | AIME25 | Avg. |
|---|---|---|---|---|---|---|
| 1024 | PS-PPO | $1.77 \pm 0.02$ | 85.0 | 16.7 | 13.3 | **38.3** |
| | S-GRPO | $2.66 \pm 0.01$ | 84.7 | 16.7 | 10.0 | 37.1 |
| | DAPO | $3.23 \pm 0.01$ | 85.2 | 16.7 | 10.0 | 37.3 |
| 2048 | PS-PPO | $2.02 \pm 0.03$ | 86.2 | 23.3 | 16.7 | **42.1** |
| | S-GRPO | $4.03 \pm 0.03$ | 85.6 | 20.0 | 13.3 | 39.6 |
| | DAPO | $4.93 \pm 0.03$ | 86.4 | 23.3 | 16.7 | **42.1** |
| 4096 | PS-PPO | $2.39 \pm 0.04$ | 86.4 | 23.3 | 16.7 | 42.1 |
| | S-GRPO | $6.70 \pm 0.04$ | 85.8 | 20.0 | 13.3 | 39.7 |
| | DAPO | $7.78 \pm 0.04$ | 86.6 | 23.3 | 16.7 | **42.2** |

ory, whereas PS-PPO achieves larger savings by truncating the gradient-update computation itself.

**Scaling with maximum completion length.** PS-PPO reduces update computation by backpropagating only through sampled prefixes, so its efficiency advantage is expected to become more pronounced as the maximum completion length grows. To test this, we vary the maximum comple-

tion length $T_{max} \in \{1024, 2048, 4096\}$ and compare PS-PPO with S-GRPO and DAPO under the same rollout and training-step budget. As shown in Table 3, PS-PPO maintains accuracy comparable to DAPO while becoming increasingly faster as $T_{max}$ grows. At $T_{max} = 4096$, PS-PPO achieves $2.8\times$ speedup over S-GRPO and $3.3\times$ speedup over DAPO in update-stage training time per step. These results show that the computational benefit of prefix sampling becomes more pronounced in the long-completion regime targeted by PS-PPO.

**Understanding the effect of cutoff strategy.** To isolate the impact of the cutoff strategy, we compare PS-PPO variants under a fixed backpropagation budget. We train all models with a maximum completion length of $T$=1024. Under the uniform-cutoff strategy, the cutoff length $H$ is sampled uniformly over timesteps, yielding an expected retained prefix length of $\mathbb{E}[H] = T/2 = 512$. We therefore set the backpropagation budget to $B$=512 so that all strategies are compared under the same expected backpropagated length.

Figure 3 plots reward versus wall-clock time. PS-PPO (Optimized) and PS-PPO (Heuristic) incur additional per-step

overhead to compute the cutoff probabilities $\xi_{1:T}(x)$, yet they reach the reward plateau substantially earlier than the baselines. This indicates that the improvement is not solely due to reduced per-step cost, but to allocating the same backpropagation budget to more informative parts of each completion, improving time-to-reward. Since the time-prior baseline does not yield clear gains over uniform cutoff, simply favoring earlier timesteps is not sufficient for effective learning.

In contrast, fixed-length truncation has low per-step overhead but can be misaligned with where useful learning signal occurs: for some prompts it truncates informative suffix tokens, while for others it propagates through uninformative suffix tokens, leading to inefficient updates. Uniform truncation exhibits the complementary issue: without guidance on which tokens to keep, some updates are cut too aggressively and discard learning signal, whereas others retain long, low-utility suffixes that increase computation with limited benefit. Overall, our optimized cutoff distribution offsets its modest overhead by extracting more learning signal per unit wall-clock time, resulting in faster training progress.

### 4.2. Analysis on Hyperparameters

**The budget size $B$.** The budget $B$ controls the expected cutoff length, which directly determines how many tokens are included in the forward and backward passes during each update. We parameterize the cutoff distribution by $\xi_t(x) := \Pr(H \geq t \mid x)$, so that $\mathbb{E}[H \mid x] = \sum_{t=1}^{T} \xi_t(x) = B$. To study its effect, we sweep $B \in \{64, 128, 256, 512\}$. Table 4 reports pass@1 on our evaluation and the training-time per step.

We observe two coupled effects as $B$ varies. With smaller budgets, truncation becomes more aggressive, which reduces training compute but also shortens the backpropagated prefix and can remove informative parts of the trajectory from the update. At the same time, smaller $B$ increases the magnitude of the truncation correction, leading to a higher variance gradient estimator. As $B$ increases, longer prefixes are retained and the correction weakens, yielding more stable optimization. However, beyond a moderate budget, accuracy gains saturate while training time continues to grow as more tokens participate in forward and backward computation. Overall, these results highlight $B$ as a trade-off parameter between computational savings and statistical efficiency, motivating our default choice of $B = 128$.

**The number of rollouts $K$.** We ablate the number of rollouts per prompt $K \in \{2, 4, 8, 16\}$ while keeping other settings fixed. In critic-free methods, the broadcast advantage takes the form $\hat{A}^{(k)} = R^{(k)} - \bar{R}$ with $\bar{R} = \frac{1}{K}\sum_{j=1}^{K} R^{(j)}$. When all rollouts for a prompt are either correct or incor-

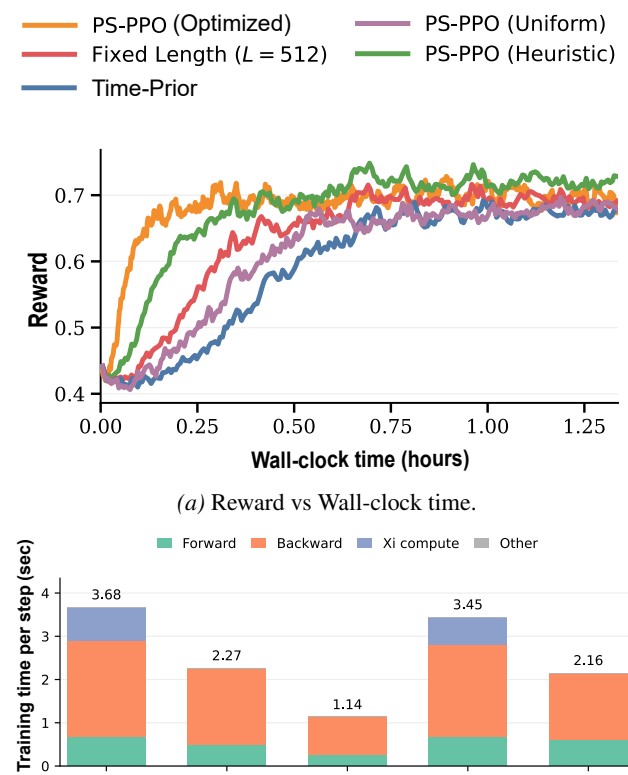

*(a)* Reward vs Wall-clock time.

*(b)* Training-time per step under different cutoff strategies.

*Figure 3.* **Understanding the effect of cutoff strategies.** (a) Reward versus wall-clock time under a matched backpropagation budget ($B$=512, $T$=1024). PS-PPO (Optimized) reaches the plateau earlier than alternative cutoff strategies. (b) Training time per step (excluding rollout/generation) with a breakdown into forward/backward, cutoff-probability ($\xi$) computation, and other costs. Prompt-conditioned cutoffs (Optimized/Heuristic) incur additional $\xi$-computation overhead, but achieve faster progress in wall-clock time.

rect, $\hat{A}^{(k)} = 0$ for all $k$, yielding no learning signal for that prompt. As $K$ increases, such degenerate cases may become less frequent, which can make training updates more stable.

This effect can be more pronounced in PS-PPO, since cutoff selection relies on a per-timestep uncertainty signal. Specifically, we use a lightweight prompt-conditioned signal $\hat{u}_t(x)$ as an upper-bound proxy for $\mathrm{Var}(R \mid s_t)$ when constructing the cutoff distribution. Estimating $\hat{u}_t(x)$ requires observing reward variability among rollouts that share similar prefixes: with small $K$, outcomes are often degenerate or too sparse, causing $\hat{u}_t(x)$ to be noisy or nearly flat across $t$, in which case the resulting cutoff distribution tends toward a conservative, near-uniform behavior. As $K$ increases, mixed outcomes tend to appear more often and the estimate of $\hat{u}_t(x)$ becomes more reliable, which yields a more informative cutoff distribution.

Empirically, increasing $K$ improves final performance but

*Table 4.* Effect of the update budget $B$ in PS-PPO (pass@1 accuracy, %). All runs use the same number of training steps and rollouts per step. All values are averaged over three independent runs.

| Benchmark | Budget $B$ | | | |
|---|---|---|---|---|
| | 64 | 128 | 256 | 512 |
| MATH500 | 82.9 | 85.0 | 85.1 | 85.0 |
| AMC23 | 54.2 | 58.5 | 57.8 | 57.5 |
| CollegeMath | 22.0 | 23.9 | 24.7 | 24.2 |
| MinervaMath | 28.0 | 30.1 | 30.4 | 30.1 |
| AIME24 | 10.0 | 16.7 | 16.7 | 16.7 |
| AIME25 | 6.7 | 13.3 | 10.0 | 13.3 |
| Avg. | 34.0 | 37.9 | 37.5 | 37.8 |
| Training-time / step (sec) | 1.15 | 1.77 | 2.01 | 2.56 |

*Table 5.* Effect of the number of rollouts $K$ in PS-PPO (pass@1 accuracy, %). All runs use the same number of training steps and the same update setting ($B$=128). Training time per step excludes rollout/generation. All values are averaged over three independent runs.

| Benchmark | #Rollouts per step $K$ | | | |
|---|---|---|---|---|
| | 2 | 4 | 8 | 16 |
| MATH500 | 82.8 | 84.0 | 85.0 | 85.2 |
| AMC23 | 55.0 | 57.5 | 58.5 | 58.1 |
| CollegeMath | 22.0 | 23.0 | 23.9 | 24.1 |
| MinervaMath | 28.7 | 29.8 | 30.1 | 30.5 |
| AIME24 | 10.0 | 13.3 | 16.7 | 20.0 |
| AIME25 | 6.7 | 10.0 | 13.3 | 13.3 |
| Avg. | 34.2 | 36.3 | 37.9 | 38.5 |
| Training time / step (sec) | 1.14 | 1.43 | 1.77 | 2.24 |

also increases training time per step, since more sampled completions induce additional training-time computation (forward/backward). In our main experiments, we use $K$=8 as a practical trade-off: it achieves performance comparable to $K$=16 while incurring substantially lower per-step training cost.

## 5. Conclusion

In this paper, we presented PS-PPO (Prefix-Sampling PPO), a framework that reduces the training cost of critic-free RLHF, where policy updates typically require backpropagation through long reasoning traces. PS-PPO derives an optimized prompt-conditioned cutoff distribution by formulating its design as a convex optimization problem, which minimizes a variance surrogate of the truncated gradient estimator under a training-budget constraint. It then samples a cutoff during training and applies an inclusion-probability correction, so that the truncated update matches the full-sequence objective in expectation while requiring substantially less computation.

Empirically, PS-PPO achieves accuracy comparable to strong critic-free baselines across mathematical reasoning benchmarks while substantially reducing training time and peak GPU memory usage. These results suggest that, in many long reasoning traces, intermediate prefixes already contain substantial information about the final outcome. By allocating update computation through stochastic prefix sampling while preserving the full-sequence update in expectation, PS-PPO offers a scalable path to aligning large language models under substantially reduced resource requirements.

## Acknowledgments

This work was supported by Institute for Information & Communications Technology Planning & Evaluation (IITP), funded by Korea government(MSIT), through the Information Technology Research Center (ITRC) program and other projects (No. RS-2022-II220311, Development of Goal-Oriented Reinforcement Learning Techniques for Contact-Rich Robotic Manipulation of Everyday Objects; No.RS-2024-00343989, Enhancing the Ethics of Data Characteristics and Generation AI Models for Social and Ethical Learning; No.RS-2024-00457882, AI Research Hub Project; No.RS-2020-II200940, Foundations of Safe Reinforcement Learning and Its Applications to Natural Language Processing; No.RS-2019-II190075, Artificial Intelligence Graduate School Program (KAIST); IITP-2026-RS-2024-00436857).

## Impact Statement

This work improves the computational efficiency of RLHF for large language models by reducing policy-update computation and peak memory. This may make post-training long-reasoning models more accessible to academic and resource-constrained groups and reduce the hardware requirements and overall cost of repeated training experiments. However, more efficient post-training may also accelerate the development of capable generative models, which can produce misleading, unsafe, biased, or harmful content if trained or deployed without appropriate safeguards. PS-PPO is an optimization method and does not by itself determine the behavior or safety properties of the resulting model. Its broader impact therefore depends on the reward specification, training data, deployment context, and safety evaluations used in conjunction with the method.

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

## A. Late-Recovery Analysis

We further examine whether final correctness often depends on information that appears only late in the completion. For each MATH-500 problem, we first generate a full greedy completion with a maximum generation length of 4096 tokens. We then take the 50% and 75% prefixes of that completion and sample 32 suffix rollouts from each prefix. Let $p_r$ denote the prefix-conditioned success rate at prefix ratio $r$.

We report two statistics. The first is prefix sufficiency, defined as the fraction of problems for which $p_r \geq 0.8$. The second is late recovery, defined as the fraction of finally correct completions whose prefix-conditioned success rate is still below $0.8$ at that prefix:

$$\Pr(p_r < 0.8 \mid \text{final correct}).$$

Table 6 shows that a large fraction of problems already have highly predictive prefixes well before the end of the completion.

*Table 6.* Prefix sufficiency and late recovery on MATH-500. Each $p_r$ is estimated from 32 suffix rollouts conditioned on the $r$-prefix of a greedy completion.

| Model | % problems $p_{0.5} \geq 0.8$ | % problems $p_{0.75} \geq 0.8$ | Late recovery @ 50% $\Pr(p_{0.5} < 0.8 \mid \text{correct})$ | Late recovery @ 75% $\Pr(p_{0.75} < 0.8 \mid \text{correct})$ |
|---|---|---|---|---|
| Qwen2.5-Math-7B | 73.4 | 82.2 | 16.9 | 7.1 |
| DeepSeek-R1-Distill-Qwen-7B | 74.2 | 84.4 | 18.9 | 8.6 |

At the 75% prefix, only 7.1% of finally correct Qwen2.5-Math-7B completions and 8.6% of finally correct DeepSeek-R1-Distill-Qwen-7B completions still have $p_{0.75} < 0.8$. Equivalently, 92.9% and 91.4% of finally correct completions already have $p_{0.75} \geq 0.8$, respectively. These results suggest that late correction can occur, but a large fraction of finally correct completions already have highly predictive prefixes before the end of the trajectory.

## B. Unbiasedness of the Truncated, Reweighted Estimator

We justify that the truncated estimator in Eq. (9) preserves the expected (full) update under the cutoff randomness. Fix a prompt $x$ and the $K$ sampled trajectories, and denote by $g_t^{(k)}(\theta)$ the per-timestep gradient contribution at timestep $t$ for trajectory $k$ that would appear in the full update

$$G(\theta) \; := \; \frac{1}{K} \sum_{k=1}^{K} \sum_{t=1}^{T} g_t^{(k)}(\theta). \tag{10}$$

Define the inclusion probability

$$\xi_t \; := \; \mathbb{P}\left(H^{(k)} \geq t \mid x\right) \in (0, 1], \tag{11}$$

and introduce the inclusion indicator $I_t^{(k)} := \mathbf{1}\{t \leq H^{(k)}\}$. Then the truncated estimator can be written as

$$\widehat{G}(\theta) = \frac{1}{K} \sum_{k=1}^{K} \sum_{t=1}^{T} \frac{I_t^{(k)}}{\xi_t} g_t^{(k)}(\theta), \tag{12}$$

where we may sum to $T$ by noting that $I_t^{(k)} = 0$ for $t > H^{(k)}$. By construction, $\mathbb{E}[I_t^{(k)} \mid x] = \mathbb{P}(H^{(k)} \geq t \mid x) = \xi_t$. Moreover, conditional on $x$ and the sampled trajectories, $g_t^{(k)}(\theta)$ is fixed and the only remaining randomness is due to the

cutoff $H^{(k)}$. Therefore,

$$\mathbb{E}\left[\widehat{G}(\theta) \mid x, o_{1:k}\right] = \frac{1}{K} \sum_{k=1}^{K} \sum_{t=1}^{T} \mathbb{E}\left[\frac{I_t^{(k)}}{\xi_t} g_t^{(k)}(\theta) \,\middle|\, x, o_{1:k}\right] \tag{13}$$

$$= \frac{1}{K} \sum_{k=1}^{K} \sum_{t=1}^{T} \frac{g_t^{(k)}(\theta)}{\xi_t} \mathbb{E}\left[I_t^{(k)} \mid x\right] \tag{14}$$

$$= \frac{1}{K} \sum_{k=1}^{K} \sum_{t=1}^{T} \frac{g_t^{(k)}(\theta)}{\xi_t} \xi_t \tag{15}$$

$$= \frac{1}{K} \sum_{k=1}^{K} \sum_{t=1}^{T} g_t^{(k)}(\theta) = G(\theta). \tag{16}$$

Finally, taking expectation over trajectory sampling yields

$$\mathbb{E}_{\tau,H}\left[\widehat{G}(\theta) \mid x\right] = \mathbb{E}_{\tau}[G(\theta) \mid x],$$

so the cutoff randomness does not change the expected full-sequence update.

## C. Derivation of Eq. (4)

Fix a prompt $x$ and omit the dependence on $x$ in $\{\xi_t\}_{t=1}^{T}$ for readability. For each completion $k \in \{1, \ldots, K\}$, let $g_t^{(k)}(\theta) \in \mathbb{R}^d$ denote the (pre-clipping) PPO contribution at timestep $t$. We sample an independent cutoff length $H^{(k)}$ with survival probabilities $\xi_t := \Pr(H^{(k)} \geq t \mid x)$. The cutoff-based (reweighted) update estimator can be written as

$$\widehat{G}(\theta) = \frac{1}{K} \sum_{k=1}^{K} \sum_{t=1}^{T} \frac{\mathbf{1}\{H^{(k)} \geq t\}}{\xi_t} g_t^{(k)}(\theta).$$

In this appendix, $\operatorname{Var}_H(\cdot \mid x)$ denotes variance over the cutoff variables $\{H^{(k)}\}_{k=1}^{K}$ only (conditioning on the realized rollouts), and we measure update variability using the trace/diagonal surrogate $\operatorname{Var}_H(\widehat{G} \mid x) \equiv \operatorname{tr}(\operatorname{Cov}_H(\widehat{G} \mid x)) = \mathbb{E}_H[\|\widehat{G} - \mathbb{E}_H[\widehat{G}]\|^2 \mid x]$.

Since $\mathbb{E}_H[\mathbf{1}\{H^{(k)} \geq t\} \mid x] = \xi_t$, we have $\mathbb{E}_H[\mathbf{1}\{H^{(k)} \geq t\}/\xi_t \mid x] = 1$, hence

$$\widehat{G}(\theta) - \mathbb{E}_H[\widehat{G}(\theta) \mid x] = \frac{1}{K} \sum_{k=1}^{K} \sum_{t=1}^{T} \left(\frac{\mathbf{1}\{H^{(k)} \geq t\}}{\xi_t} - 1\right) g_t^{(k)}(\theta).$$

Because cutoffs are sampled independently across $k$, cross-completion covariances vanish under $\operatorname{Var}_H(\cdot \mid x)$, and thus

$$\operatorname{Var}_H(\widehat{G}(\theta) \mid x) = \frac{1}{K} \mathbb{E}_H\left[\left\|\sum_{t=1}^{T} \left(\frac{\mathbf{1}\{H \geq t\}}{\xi_t} - 1\right) g_t(\theta)\right\|^2 \,\middle|\, x\right],$$

where $(H, g_t)$ denotes a generic completion (dropping the superscript $k$). Expanding the squared norm yields diagonal terms and cross-timestep inner products; under the diagonal/trace surrogate we ignore the cross-timestep covariances and keep only the diagonal terms:

$$\operatorname{Var}_H(\widehat{G}(\theta) \mid x) \approx \frac{1}{K} \sum_{t=1}^{T} \mathbb{E}_H\left[\left(\frac{\mathbf{1}\{H \geq t\}}{\xi_t} - 1\right)^2 \,\middle|\, x\right] \|g_t(\theta)\|^2.$$

For each $t$, since $\Pr(H \geq t \mid x) = \xi_t$,

$$\mathbb{E}_H\left[\left(\frac{\mathbf{1}\{H \geq t\}}{\xi_t} - 1\right)^2 \,\middle|\, x\right] = \xi_t \left(\frac{1}{\xi_t} - 1\right)^2 + (1 - \xi_t) \cdot 1 = \frac{1}{\xi_t} - 1.$$

Substituting gives

$$\text{Var}_H(\widehat{G}(\theta) \mid x) \approx \frac{1}{K} \sum_{t=1}^{T} \left(\frac{1}{\xi_t} - 1\right) \|g_t(\theta)\|^2.$$

Finally, taking expectation over rollouts conditioned on $x$ and defining $w_t(x) := \mathbb{E}[\|g_t^{(k)}(\theta)\|^2 \mid x]$ yields

$$\mathbb{E}\left[\text{Var}_H(\widehat{G}(\theta) \mid x)\right] \approx \frac{1}{K} \sum_{t=1}^{T} w_t(x) \left(\frac{1}{\xi_t} - 1\right),$$

which is Eq. (4).

## D. Output-Head Score Proxy

We derive a forward-only proxy for the per-token score-norm factor that avoids additional backward passes. For an autoregressive LM with a softmax output head at step $t$,

$$z_t = Wh_t + b, \qquad p_t = \mathrm{softmax}(z_t), \qquad o_t \sim p_t,$$

where $h_t$ is the last-layer hidden state (available from the forward pass) and $W, b$ are the output-head parameters.

For the sampled token $o_t$, the score with respect to the output-head parameters admits a closed form:

$$\nabla_W \log p_t(o_t) = (e_{o_t} - p_t)\, h_t^\top, \qquad \nabla_b \log p_t(o_t) = e_{o_t} - p_t.$$

Consequently,

$$\left\| \nabla_W \log p_t(o_t) \right\|_F^2 = \|h_t\|_2^2 \, \|e_{o_t} - p_t\|_2^2, \qquad \left\| \nabla_b \log p_t(o_t) \right\|_2^2 = \|e_{o_t} - p_t\|_2^2,$$

and the probability-only term satisfies

$$\|e_{o_t} - p_t\|_2^2 = 1 - 2p_t(o_t) + \|p_t\|_2^2.$$

Thus, the output-head contribution to the score norm is determined entirely by forward-pass quantities $(h_t, p_t)$.

In the main text, we use the following proxy (corresponding to the weight-matrix contribution):

$$\gamma_t \;:=\; \|h_t\|_2^2 \Big( 1 - 2p_t(o_t) + \|p_t\|_2^2 \Big).$$

Since $(W, b)$ is a subset of all parameters, the full score norm satisfies

$$\left\| \nabla_\theta \log \pi_\theta(o_t \mid s_t) \right\|_2^2 \;\geq\; \left\| \nabla_{W,b} \log p_t(o_t) \right\|_2^2,$$

because squared norms decompose additively across parameter blocks. Finally, we incorporate this proxy into the timestep weights by defining

$$\tilde{w}_t(x) := \mathbb{E}\!\left[ \hat{A}_t^2 \, \gamma_t \,\middle|\, x \right],$$

where the expectation is over on-policy rollouts generated by $\pi_{\theta_{\mathrm{old}}}(\cdot \mid x)$.

A practical concern is that $\|p_t\|_2^2 = \sum_i p_{t,i}^2$ appears to require the full vocabulary distribution. Let $z_t \in \mathbb{R}^{|V|}$ be the logits and define two log-sum-exp reductions:

$$\ell_1 := \log \sum_i e^{z_{t,i}}, \qquad \ell_2 := \log \sum_i e^{2z_{t,i}}.$$

Then

$$\|p_t\|_2^2 = \sum_i \left( \frac{e^{z_{t,i}}}{\sum_j e^{z_{t,j}}} \right)^2 = \frac{\sum_i e^{2z_{t,i}}}{\left( \sum_j e^{z_{t,j}} \right)^2} = \exp(\ell_2 - 2\ell_1).$$

Moreover, $p_t(o_t) = \exp(z_{t,o_t} - \ell_1)$ is already available when computing the token log-probability. Therefore, $\gamma_t$ can be computed using only forward activations $(h_t)$ and two scalar reductions $(\ell_1, \ell_2)$, without any additional backward passes.

# E. Derivation of the Reward-Uncertainty Upper Bound

We derive the reward-uncertainty proxy used in Eq. (7). Consider a binary terminal reward $R \in \{0, 1\}$. For a prompt $x$, let $S_t$ denote the random prefix state at timestep $t$, and define

$$p_t(s_t) := \Pr(R = 1 \mid S_t = s_t), \qquad p(x) := \Pr(R = 1 \mid x).$$

Since $R$ is Bernoulli conditioned on $S_t = s_t$,

$$\mathrm{Var}(R \mid S_t = s_t) = p_t(s_t)(1 - p_t(s_t)) \leq \min\{p_t(s_t), 1 - p_t(s_t)\}. \tag{17}$$

Using

$$\min\{a, b\} = \frac{1}{2}(a + b - |a - b|)$$

with $a = p_t(s_t)$ and $b = 1 - p_t(s_t)$, we obtain

$$\min\{p_t(s_t), 1 - p_t(s_t)\} = \frac{1}{2} - \frac{1}{2}|2p_t(s_t) - 1|. \tag{18}$$

Taking expectation over $S_t \mid x$ gives

$$\mathbb{E}[\mathrm{Var}(R \mid S_t) \mid x] \leq \frac{1}{2} - \frac{1}{2}\mathbb{E}[|2p_t(S_t) - 1| \mid x]. \tag{19}$$

We now rewrite the absolute-value term. Since

$$2p_t(s_t) - 1 = \Pr(R = 1 \mid S_t = s_t) - \Pr(R = 0 \mid S_t = s_t),$$

we have

$$\mathbb{E}[|2p_t(S_t) - 1| \mid x] = \sum_{s_t} \Pr(s_t \mid x) \left|\Pr(R = 1 \mid s_t) - \Pr(R = 0 \mid s_t)\right|$$
$$= \sum_{s_t} |\Pr(s_t, R = 1 \mid x) - \Pr(s_t, R = 0 \mid x)|. \tag{20}$$

Substituting Eq. (20) into Eq. (19) yields

$$\mathbb{E}[\mathrm{Var}(R \mid S_t) \mid x] \leq \frac{1}{2} - \frac{1}{2} \left\|\Pr(S_t, R = 1 \mid x) - \Pr(S_t, R = 0 \mid x)\right\|_1. \tag{21}$$

The state distribution in Eq. (21) is not directly convenient to estimate during training. We therefore map prefix states to their next-token predictive distributions under the old policy. Define

$$\phi_t(s_t) := \Pr(S_t = s_t, R = 1 \mid x) - \Pr(S_t = s_t, R = 0 \mid x). \tag{22}$$

Because $\sum_{o_t \in \mathcal{V}} \pi_{\theta_{\mathrm{old}}}(o_t \mid s_t) = 1$, the triangle inequality gives

$$\sum_{s_t} |\phi_t(s_t)| = \sum_{o_t \in \mathcal{V}} \sum_{s_t} |\phi_t(s_t)| \pi_{\theta_{\mathrm{old}}}(o_t \mid s_t)$$
$$\geq \sum_{o_t \in \mathcal{V}} \left|\sum_{s_t} \phi_t(s_t) \pi_{\theta_{\mathrm{old}}}(o_t \mid s_t)\right|. \tag{23}$$

Combining Eq. (21) and Eq. (23) gives the looser but token-distribution-based upper bound

$$\mathbb{E}[\mathrm{Var}(R \mid S_t) \mid x] \leq \frac{1}{2} - \frac{1}{2} \sum_{o_t \in \mathcal{V}} \left|\sum_{s_t} \phi_t(s_t) \pi_{\theta_{\mathrm{old}}}(o_t \mid s_t)\right|. \tag{24}$$

Next define the state-averaged next-token distributions over successful and unsuccessful rollouts:

$$\bar{\pi}_G(\cdot \mid t, x) := \mathbb{E}[\pi_{\theta_{\text{old}}}(\cdot \mid S_t) \mid x, R = 1],$$

$$\bar{\pi}_B(\cdot \mid t, x) := \mathbb{E}[\pi_{\theta_{\text{old}}}(\cdot \mid S_t) \mid x, R = 0].$$

Using $\Pr(S_t = s_t, R = r \mid x) = \Pr(R = r \mid x)\Pr(S_t = s_t \mid x, R = r)$, we obtain, for each token $o_t$,

$$\sum_{s_t} \phi_t(s_t)\pi_{\theta_{\text{old}}}(o_t \mid s_t) = p(x)\bar{\pi}_G(o_t \mid t, x) - (1 - p(x))\bar{\pi}_B(o_t \mid t, x). \tag{25}$$

Substituting Eq. (25) into Eq. (24), we define the reward-uncertainty proxy

$$u_t(x) := \frac{1}{2} - \frac{1}{2} \left\| p(x)\bar{\pi}_G(\cdot \mid t, x) - (1 - p(x))\bar{\pi}_B(\cdot \mid t, x) \right\|_1, \tag{26}$$

which satisfies

$$\mathbb{E}[\text{Var}(R \mid S_t) \mid x] \le u_t(x). \tag{27}$$

In practice, $p(x)$, $\bar{\pi}_G$, and $\bar{\pi}_B$ are estimated from the $K$ completions sampled for the same prompt. This yields the empirical quantity $\hat{u}_t(x)$, which is used as the reward-uncertainty term in the cutoff-design objective.

## F. Monotone Budget Design and PAV

We design monotone inclusion probabilities $\xi_{1:T}$ under an expected-update budget $B$:

$$\min_{\xi_{1:T}} \sum_{t=1}^T \frac{w_t}{\xi_t} \quad \text{s.t.} \quad \sum_{t=1}^T \xi_t = B, \ 1 \ge \xi_1 \ge \cdots \ge \xi_T > 0, \tag{28}$$

where $w_t \ge 0$ is the timestep weight and $\xi_t = \Pr(t \le H)$.

**Unconstrained solution (ignoring monotonicity).** First ignore the monotonicity constraints and consider

$$\min_{\xi_t > 0} \sum_{t=1}^T \frac{w_t}{\xi_t} \quad \text{s.t.} \quad \sum_{t=1}^T \xi_t = B. \tag{29}$$

The Lagrangian is

$$\mathcal{L}(\xi, \lambda) = \sum_{t=1}^T \frac{w_t}{\xi_t} + \lambda\Big(\sum_{t=1}^T \xi_t - B\Big).$$

For any $t$ with $\xi_t > 0$, the stationarity condition gives

$$\frac{\partial \mathcal{L}}{\partial \xi_t} = -\frac{w_t}{\xi_t^2} + \lambda = 0 \quad \Longrightarrow \quad \xi_t = \sqrt{\frac{w_t}{\lambda}}.$$

Using the budget constraint,

$$\sum_{t=1}^T \sqrt{\frac{w_t}{\lambda}} = B \quad \Longrightarrow \quad \frac{1}{\sqrt{\lambda}} = \frac{B}{\sum_{j=1}^T \sqrt{w_j}},$$

so the closed form is

$$\xi_t^\star = B \cdot \frac{\sqrt{w_t}}{\sum_{j=1}^T \sqrt{w_j}}. \tag{30}$$

(If one also enforces $\xi_t \le 1$, the KKT conditions imply a prefix of saturated coordinates $\xi_t = 1$ may occur; the same derivation applies to the remaining suffix with a reduced budget.)

**Blockwise form under monotonicity.** Under the monotonicity constraint $\xi_1 \geq \cdots \geq \xi_T$, the KKT conditions imply that the optimal solution is piecewise constant over contiguous blocks. Let $\{[i_m, j_m]\}_{m=1}^M$ be a partition of $\{1, \ldots, T\}$ into contiguous intervals, and define the block length $L_m := j_m - i_m + 1$ and the aggregated weight $W_m := \sum_{t=i_m}^{j_m} w_t$. Restricting to blockwise-constant variables $\xi_t = \xi_m$ for $t \in [i_m, j_m]$ yields the reduced program

$$\min_{\xi_{1:M}} \sum_{m=1}^M \frac{W_m}{\xi_m} \quad \text{s.t.} \quad \sum_{m=1}^M L_m \xi_m = B, \ \xi_1 \geq \cdots \geq \xi_M > 0. \tag{31}$$

**Closed form for a fixed partition.** For a *fixed* block partition, temporarily drop the inter-block monotonicity in (31) and solve

$$\min_{\xi_m > 0} \sum_{m=1}^M \frac{W_m}{\xi_m} \quad \text{s.t.} \quad \sum_{m=1}^M L_m \xi_m = B.$$

The Lagrangian is

$$\mathcal{L}(\xi, \lambda) = \sum_{m=1}^M \frac{W_m}{\xi_m} + \lambda \Big( \sum_{m=1}^M L_m \xi_m - B \Big),$$

and stationarity gives

$$-\frac{W_m}{\xi_m^2} + \lambda L_m = 0 \quad \Longrightarrow \quad \xi_m = \sqrt{\frac{W_m}{\lambda L_m}}.$$

Imposing the budget constraint,

$$\sum_{m=1}^M \sqrt{\frac{L_m W_m}{\lambda}} = B \quad \Longrightarrow \quad \frac{1}{\sqrt{\lambda}} = \frac{B}{\sum_{j=1}^M \sqrt{L_j W_j}}.$$

Therefore the blockwise optimum for the fixed partition is

$$\xi_m^\star = B \cdot \frac{\sqrt{W_m/L_m}}{\sum_{j=1}^M \sqrt{L_j W_j}} = B \cdot \frac{s_m}{\sum_{j=1}^M L_j s_j}, \qquad s_m := \sqrt{W_m/L_m}. \tag{32}$$

When the block scores satisfy $s_1 \geq \cdots \geq s_M$, this solution already obeys the monotonicity constraint; otherwise, PAV merges adjacent violating blocks until the scores become nonincreasing, yielding the globally optimal monotone solution.

# G. Text Generation Tasks with Continuous Rewards

We evaluate PS-PPO and baselines on text generation tasks with continuous scalar rewards produced by learned reward models. We use Qwen2.5-3B-Instruct[3] and Llama-3.1-8B-Instruct[4] as base models. For Positive Generation, prompts are drawn from the IMDB dataset (Maas et al., 2011) and rewards are computed using lvwerra/gpt2-imdb[5]. For Helpful Assistant, prompts are drawn from HH-RLHF (Bai et al., 2022) and rewards are computed using weqweasdas/RM-Gemma-2B[6].

To reuse the same $u_t(x)$ estimator under continuous rewards, we stochastically convert the scalar reward $r$ into a binary variable only for $u_t(x)$ estimation. Specifically, we map $r$ to a Bernoulli probability via a sigmoid:

$$p = \sigma(r) = \frac{1}{1 + \exp(-r)},$$

and sample a binary label $\tilde{y} \sim \text{Bernoulli}(p)$. We use $\tilde{y} \in \{0, 1\}$ only inside $u_t(x)$ estimation, while the policy update still uses the original continuous reward $r$.

*Table 7.* Results on generation tasks measured by average reward. All values are averaged over three independent runs. (**Bold**: best, underlined: second best).

| Base model | Method | Helpful assistant | Positive Generation | Avg. |
|---|---|---|---|---|
| Qwen2.5-3B-Instruct | Base | $4.08 \pm 0.06$ | $5.74 \pm 0.08$ | $4.93 \pm 0.05$ |
| | GRPO | $5.29 \pm 0.08$ | $\underline{7.58 \pm 0.10}$ | $6.43 \pm 0.07$ |
| | Dr.GRPO | $5.61 \pm 0.09$ | $7.14 \pm 0.11$ | $6.35 \pm 0.08$ |
| | RLOO | $4.89 \pm 0.07$ | $7.00 \pm 0.09$ | $5.93 \pm 0.06$ |
| | DAPO | $\mathbf{6.18 \pm 0.07}$ | $\mathbf{7.97 \pm 0.08}$ | $\mathbf{7.07 \pm 0.06}$ |
| | DAPO (w/ forking tokens) | $5.95 \pm 0.08$ | $7.46 \pm 0.09$ | $6.71 \pm 0.07$ |
| | S-GRPO | $5.11 \pm 0.10$ | $6.73 \pm 0.12$ | $5.90 \pm 0.08$ |
| | PS-PPO (Ours) | $\underline{6.02 \pm 0.08}$ | $7.42 \pm 0.09$ | $\underline{6.72 \pm 0.07}$ |
| Llama-3.1-8B-Instruct | Base | $4.71 \pm 0.05$ | $6.44 \pm 0.06$ | $5.59 \pm 0.05$ |
| | GRPO | $6.36 \pm 0.07$ | $8.16 \pm 0.08$ | $7.26 \pm 0.07$ |
| | Dr.GRPO | $6.82 \pm 0.08$ | $\underline{8.75 \pm 0.07}$ | $7.79 \pm 0.06$ |
| | RLOO | $5.94 \pm 0.06$ | $7.71 \pm 0.09$ | $6.81 \pm 0.07$ |
| | DAPO | $\mathbf{7.12 \pm 0.06}$ | $\mathbf{9.13 \pm 0.05}$ | $\mathbf{8.13 \pm 0.04}$ |
| | DAPO (w/ forking tokens) | $6.90 \pm 0.07$ | $8.65 \pm 0.07$ | $7.78 \pm 0.05$ |
| | S-GRPO | $6.14 \pm 0.07$ | $7.99 \pm 0.08$ | $7.09 \pm 0.06$ |
| | PS-PPO (Ours) | $\underline{7.00 \pm 0.06}$ | $8.69 \pm 0.06$ | $\underline{7.85 \pm 0.05}$ |

*Table 8.* Efficiency–performance trade-off on continuous-reward text generation tasks. Total training time and peak GPU memory are reported together with average reward. All values are averaged over three independent runs.

| Base model | Method | Total training time | Peak GPU memory | Avg. reward |
|---|---|---|---|---|
| Qwen2.5-3B-Instruct | PS-PPO (Ours) | $3.2 \pm 0.1$ h | $42.8 \pm 0.7$ GB | $6.72 \pm 0.07$ |
| | S-GRPO | $6.7 \pm 0.1$ h | $68.9 \pm 0.9$ GB | $5.90 \pm 0.08$ |
| | DAPO | $7.3 \pm 0.1$ h | $73.8 \pm 1.0$ GB | $7.07 \pm 0.06$ |
| Llama-3.1-8B-Instruct | PS-PPO (Ours) | $4.4 \pm 0.1$ h | $45.7 \pm 0.9$ GB | $7.85 \pm 0.05$ |
| | S-GRPO | $8.1 \pm 0.2$ h | $72.6 \pm 1.0$ GB | $7.09 \pm 0.06$ |
| | DAPO | $11.3 \pm 0.2$ h | $77.9 \pm 1.1$ GB | $8.13 \pm 0.04$ |

Tables 7 and 8 show that PS-PPO maintains competitive reward performance on continuous-reward generation tasks while reducing total training time and peak GPU memory. These results suggest that PS-PPO is not limited to sparse binary rewards, since the stochastic truncation estimator preserves the corresponding full-sequence update in expectation under general terminal rewards.

---

[3] https://huggingface.co/Qwen/Qwen2.5-3B-Instruct
[4] https://huggingface.co/meta-llama/Llama-3.1-8B-Instruct
[5] https://huggingface.co/lvwerra/gpt2-imdb
[6] https://huggingface.co/weqweasdas/RM-Gemma-2B

# H. Output-Head Score Norm as a Proxy for the Full Score Norm

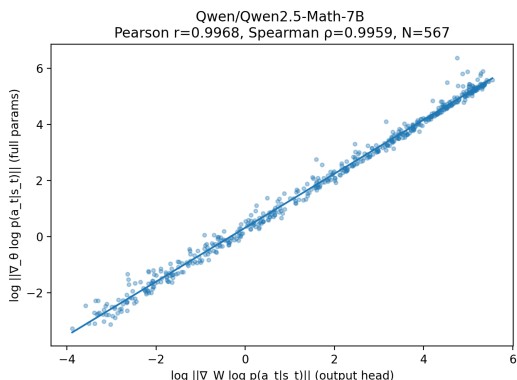

*Figure 4.* Token-wise correlation between the output-head score norm $\|\nabla_{\theta_{\text{out}}} \log \pi_\theta(o_t \mid s_t)\|$ (x-axis) and the full score norm $\|\nabla_\theta \log \pi_\theta(o_t \mid s_t)\|$ (y-axis) on Qwen2.5-Math-7B (log–log scale). We plot tokens with numerically non-negligible score norms for visual clarity. The strong correlation supports using the output-head term as a lightweight proxy.

# I. Implementation Details for Training

**Training setup.**   For mathematical reasoning experiments, we trained the 7B-scale models on 8 NVIDIA A100 GPUs. With this configuration, training on the MATH dataset took roughly 4 hours for the 7B models and 5 hours for the 8B models. All methods were implemented on top of the Hugging Face Open-R1 codebase.[7] The project repository is available at https://github.com/doohwan383/PS-PPO.

**Hyperparameters.**   We used a per-device batch size of 16, truncating input prompts to 1024 tokens and generating up to 1024 tokens per rollout during training. The learning rate was set to $5 \times 10^{-5}$, and PPO clipping used $\epsilon = 0.1$. We trained for up to 3 epochs and sampled 8 completions per prompt during rollout. Unless stated otherwise, we set the cutoff budget parameter B to 128 and compute $u_t(x)$ using a top-$k$ approximation with $k = 16$.

**Reward and evaluation.**   We followed the Qwen-Math prompting template and used a binary accuracy reward: a completion receives reward 1 if its final answer exactly matches the ground-truth answer, and 0 otherwise. For evaluation across all benchmarks, we used a maximum sequence length of 4096 tokens and reported zero-shot pass@1 accuracy. Metrics were computed using Lighteval[8] and the official Qwen2.5-Math evaluation code.[9]

---

[7]https://github.com/huggingface/open-r1
[8]https://github.com/huggingface/lighteval
[9]https://github.com/QwenLM/Qwen2.5-Math

