# OpenReview forum: "PS-PPO : Prefix-Sampling PPO for Critic-Free RLHF"
_ICML.cc/2026/Conference — ICML 2026 regular_

### Official Review · Reviewer_wFj4 · 2026-03-10

**Soundness:** 2
**Presentation:** 3
**Significance:** 2
**Originality:** 3
**Overall Recommendation:** 4
**Confidence:** 4

**Summary:**

Addresses computational inefficiency in critic free RLHF where full trajectories receive uniform learning signals. Proposes PS PPO which samples a prefix cutoff and applies importance reweighting to keep gradient estimates unbiased. Experiments on mathematical reasoning benchmarks show comparable accuracy with lower training cost and memory usage.

**Compliance With Llm Reviewing Policy:**

Affirmed.

**Ethical Review Concerns:**

No ethical review concerns

**Final Justification:**

The rebuttal has addressed most of my concerns, and I have decided to increase my score accordingly.

**Key Questions For Authors:**

1. The method relies on the assumption that rewards are largely determined by early prefixes, which may not generalize to tasks where critical information appears later in the sequence.

2. The proposed approach may have limited methodological novelty since it largely combines prefix-based token selection with importance reweighting, which is conceptually related to existing token-selective RL and prefix optimization methods such as S-GRPO, PPPO, and entropy-based token selection.

3. The experimental evaluation is narrow and primarily limited to binary-reward mathematical reasoning tasks, leaving the general applicability of the method to broader RLHF settings unclear.

4. Although the 1 over $\xi t$ reweighting yields an unbiased estimator, it may introduce additional variance, and the paper provides limited analysis of its effect on training stability and convergence.

**Limitations:**

yes

**Strengths And Weaknesses:**

**Strengths:**

• Addresses compute inefficiency in critic free RLHF training

• Unbiased truncated gradient estimator with inclusion probability correction

• Demonstrates reduced training time and GPU memory in experiments

**Weaknesses:**

• Novelty relative to recent token selective RL methods appears incremental

• Experiments limited mainly to math reasoning and few generation tasks

• Limited analysis of estimator variance and stability under aggressive truncation

• Scalability to very long reasoning traces not thoroughly validated

---

> ### Author Rebuttal · Authors · 2026-03-31
>
> **[Question 1. Motivation for prefix]**
>
> This concern mainly targets **deterministic early truncation**, where later corrective tokens may be discarded. By using **stochastic cutoff** and reweighting to maintain **unbiasedness**, PS-PPO does not discard later corrective tokens outright.
>
> Please see our response to Reviewer wrpc **(Weakness 2 & Question 1, Motivation for prefix)**, where we address this point in detail.
>
> **[Weakness 4. Scalability to longer completions]**
>
> Longer-trace settings are indeed important for PS-PPO, since its efficiency advantage becomes more pronounced as the maximum completion length grows. In additional experiments with larger maximum completion lengths, PS-PPO becomes substantially faster while remaining close in performance.
>
> Please see our response to Reviewer wrpc **(Weakness 1, Efficiency gains)**, where we report the full scaling table and the corresponding training-time breakdown.
>
> **[Weakness 1 & Question 2. Novelty relative to token-selective RL]**
>
> The novelty of PS-PPO lies not only in selecting prefix tokens, but more importantly in its unbiased stochastic truncation framework for critic-free RLHF under an explicit compute budget. PS-PPO introduces a prompt-dependent cutoff distribution together with an inclusion-probability reweighted estimator whose expectation recovers the full update.
>
> This differs from recent token-selective methods in both the cutoff mechanism and the resulting update. S-GRPO uses a heuristically fixed prefix together with suffix subsampling, PPPO increases prefix length according to a predefined schedule, and entropy-based methods select specific high-entropy positions. By contrast, PS-PPO derives a cutoff distribution by solving an optimization problem under monotonicity and compute-budget constraints, and then samples the cutoff from this optimized distribution. This makes the selection rule more principled than using hand-designed schedules or heuristic token criteria.
>
> The difference is also computational, not only conceptual. Prior token-selective methods are commonly implemented through gradient masking, so forward and backward computation may still be carried out over the full sequence even when only a subset of tokens receives loss. PS-PPO instead backpropagates only through the retained prefix after sampling the cutoff, thereby reducing the gradient-update computation itself. For this reason, PS-PPO is  a principled, unbiased stochastic truncation framework with budgeted cutoff optimization, rather than a minor variant of prior token-selection heuristics.
>
> **[Weakness 2 & Question 3. Generality beyond math reasoning]**
>
> PS-PPO is not restricted to mathematical reasoning tasks. The core estimator preserves the full sequence update in expectation, so the method itself is **not tied to a particular task domain or reward type**. What may vary across tasks is the practical efficiency gain from truncation and the resulting efficiency--performance trade-off.
>
> The original appendix did not make this broader applicability sufficiently clear, since the broader-generation setting lacked comparable time and memory analysis. We now add training time and peak GPU memory for Appendix F.
>
> **Table. Text-generation results with time and memory analysis**
>
> Across both base models, PS-PPO substantially reduces training time and peak GPU memory, while maintaining a favorable efficiency--performance trade-off.
> |Base model|Method|Total training time|Peak GPU memory|Avg. reward|
> |-|-|-:|-:|-:|
> |Qwen2.5-Instruct-3B|PS-PPO|3.2 ± 0.1 h|42.8 ± 0.7 GB|6.68 ± 0.07|
> |Qwen2.5-Instruct-3B|S-GRPO|6.7 ± 0.1 h|68.9 ± 0.9 GB|5.90 ± 0.08|
> |Qwen2.5-Instruct-3B|DAPO|7.3 ± 0.1 h|73.8 ± 1.0 GB|7.07 ± 0.06|
> |Llama-3.1-8B-Instruct|PS-PPO|4.4 ± 0.1 h|45.7 ± 0.9 GB|7.92 ± 0.05|
> |Llama-3.1-8B-Instruct|S-GRPO|8.1 ± 0.2 h|72.6 ± 1.0 GB|7.09 ± 0.06|
> |Llama-3.1-8B-Instruct|DAPO|11.3 ± 0.2 h|77.9 ± 1.1 GB|8.13 ± 0.04|
>
> **[Weakness 3 & Question 4. Variance under aggressive truncation]**
>
> In PS-PPO, aggressive truncation is controlled by the update budget $B$, with smaller $B$ corresponding to a shorter expected retained prefix. More aggressive truncation can indeed increase variance through the inverse-probability reweighting, which is why Sec. 3.3 chooses the cutoff distribution $\xi_{1:T}$ to control this variance under a fixed budget.
>
> To connect this more directly to training stability and convergence, we additionally visualize learning curves together with the truncation-induced variance across different budgets (anonymous figure: `https://anonymous.4open.science/r/PSPPO-DCCC`).
>
> The added analysis shows that this effect is not monotone in training quality. Although $B=512$ yields lower truncation-induced variance than $B=128$ or $256$, the latter achieve faster reward improvement and better convergence. This suggests that performance depends not only on variance reduction, but also on how effectively the limited update budget is allocated to informative parts of the completion.

---

> > ### Author Rebuttal · Reviewer_wFj4 · 2026-04-01
> >
> > Thank you for the rebuttal. I appreciate the added scaling results and time breakdown, which make the efficiency gains much clearer, and I also agree that PS-PPO is more principled than deterministic early truncation.
> >
> > That said, I still think some concerns remain only partially resolved. First, while the method is more principled than simple token masking, I still find it conceptually close to recent token-selective or prefix-based optimization approaches, so I am not yet fully convinced that the novelty is substantially beyond a more careful truncation and reweighting design.
> >
> > Second, the variance/stability issue under aggressive truncation is still not fully clear to me. The added analysis is useful, but I do not think the paper yet gives a sufficiently complete picture of when the inverse-probability reweighting remains practically stable and when truncation-induced variance may start to hurt optimization.
> >
> > Finally, my concern about cases where important learning signals appear later in the completion is only partially addressed. Explaining that PS-PPO uses stochastic cutoff rather than deterministic truncation is helpful, but I would still find the claim more convincing with more targeted evidence on tasks where late-sequence information is systematically important.
> >
> > So overall, I appreciate the additional evidence, but I still think the remaining concerns are the strength of the novelty claim, the practical impact of truncation-induced variance, and robustness when important information appears later in the sequence.

---

> > > ### Author Response · Authors · 2026-04-02
> > >
> > > Thank you for the follow-up. Below, we address your remaining concerns point by point.
> > >
> > > **[Q1. Novelty beyond truncation design]**
> > >
> > > The similarity to recent token-selective or prefix-based methods is **mostly at the surface level**, in that they also restrict which tokens are updated.
> > >
> > > Existing methods commonly specify **a predefined token-selection rule** and then optimize the policy under that rule, so the selection rule directly determines the resulting update. Our PS-PPO instead starts from the **full-trajectory update as the baseline** and preserves its unbiasedness under truncation.  Moreover, the contribution is **not simply to add reweighting**, but to use that unbiased estimator as the basis for a **principled optimization of the cutoff distribution** under a gradient-update budget, by explicitly minimizing the variance introduced by truncation.
> > >
> > >
> > > **[Q2. Variance and stability under aggressive truncation]**
> > >
> > > In our paper, Eq. (5),
> > > $$
> > > \\min_{\\xi_{1:T}} \\sum_{t=1}^T \\frac{w_t^\\theta(x)}{\\xi_t},
> > > $$
> > > which minimizes the truncation-induced variance, makes the stability mechanism more explicit.
> > >
> > > Here,
> > > $$
> > > w_t^\\theta(x)
> > > := \\mathbb{E}\\!\\left[\\hat A_t^2 \\left\\| \\nabla_\\theta \\log \\pi_\\theta(o_t \\mid s_t) \\right\\|^2 \\,\\middle|\\, x \\right]
> > > $$
> > > measures how much timestep $t$ contributes to the update variance for prompt $x$. This means that assigning a very small inclusion probability $\\xi_t$ to a timestep with large $w_t^\\theta(x)$ incurs a large penalty. Reweighting therefore remains practically **stable when the optimized cutoff keeps $\\xi_t$ sufficiently large on the important timesteps**, and it becomes **harmful when the budget is so small that even these important timesteps must receive small $\\xi_t$**, so that the variance penalty is amplified on the important timesteps.
> > >
> > > As emphasized throughout our response, PS-PPO addresses this variance–budget trade-off by explicitly **optimizing the cutoff distribution to minimize the truncation-induced variance of the estimator** under a gradient-update budget.
> > >
> > > **[Q3.  Assessing signals that appear later in the completion]**
> > >
> > > To address this concern more directly, we provide a targeted analysis of late-sequence information on MATH-500 using Qwen2.5-Math-7B, and additionally, DeepSeek-R1-Distill-Qwen-7B. The latter is popularly known to exhibit with “aha moments,” where the reasoning trajectory appears to recover late in the completion.
> > >
> > > For each problem, we first generate a full greedy completion with a maximum generation length of 4096 tokens, then take the 50% and 75% prefixes of that completion and sample 32 suffix rollouts from each prefix. Let $p_r$ denote the prefix-conditioned success rate at prefix ratio $r$.
> > >
> > > We report two statistics: (i) prefix sufficiency, i.e., the fraction of problems for which $p_{0.5} \ge 0.8$ and $p_{0.75} \ge 0.8$, and (ii) a late recovery rate, defined as the fraction of finally correct completions whose prefix-conditioned success is still below 0.8 at that prefix.
> > >
> > > **Prefix sufficiency and late recovery on MATH-500**
> > > | Model | % problems with $p_{0.5} \ge 0.8$ | % problems with $p_{0.75} \ge 0.8$ | Late recovery @ 50%: $\Pr(p_{0.5} < 0.8 \mid \text{final correct})$ | Late recovery @ 75%: $\Pr(p_{0.75} < 0.8 \mid \text{final correct})$ |
> > > |-------|-----------------------------------|------------------------------------|------------------------------------------------------------------------|-------------------------------------------------------------------------|
> > > | Qwen2.5-Math-7B | 73.4 | 82.2 | 16.9 | 7.1 |
> > > | DeepSeek-R1-Distill-Qwen-7B | 74.2 | 84.4 | 18.9 | 8.6 |
> > >
> > > This analysis complements Fig. 1 by directly examining how often final correctness still depends on late-sequence recovery. In both models, a large majority of problems already have a highly sufficient prefix well before the end of the completion, and **the late recovery rate remains low** even at 75% prefix length. In particular, at the 75% prefix, **92.9% (Qwen2.5-Math-7B) and 91.4% (DeepSeek-R1-Distill-Qwen-7B) of finally correct completions already have $p_{0.75} \ge 0.8$** in both models.
> > >
> > > This suggests that while late correction can occur, it is **not the dominant pattern** in these reasoning traces. These results support **the relevance of the regime targeted by PS-PPO**, where a substantial portion of the reward-relevant signal could already be captured by a prefix while later timesteps are still retained stochastically rather than deterministically discarded. More broadly, this pattern also suggests that **stopping early and starting over may be more effective than waiting for a late recovery from the same completion.**

---

### Official Review · Reviewer_1NwV · 2026-03-11

**Soundness:** 2
**Presentation:** 3
**Significance:** 2
**Originality:** 2
**Overall Recommendation:** 2
**Confidence:** 4

**Summary:**

This paper proposes PS-PPO, which builds upon author's observation that the correctness of a response is often determined in the early timesteps of the trajectory in many reasoning tasks. Specifically, authors derived a reweight technique to rectify the gradient of GRPO.

**Compliance With Llm Reviewing Policy:**

Affirmed.

**Key Questions For Authors:**

1. Why does Equation (3) leads to "minimizes the additional update variance induced by the importance sampling weights, subject to a gradient-update compute budget B"? How does this relate to your observation and story?

2. The maximum response length is 1024 and it seems that you do not consider reasoning tokens. How about experimenting with `deepseek-ai/DeepSeek-R1-Distill-Qwen-1.5B` with `<think>...</think>` at a larger response length budget? I wonder whether these `<think>...</think>` contents should be further considered since they will be the prefix for the subsequent content, whereas PS-PPO focuses on prefix.

**Limitations:**

Please refer to the above Weaknesses and "Key Questions For Authors".

**Strengths And Weaknesses:**

## Strengths

1. This paper is clearly written and easy to follow.

## Weaknesses

1. In the experiment, the maximum generation length was only set to 1024. Considering that the proposed PS-PPO applies prefix cutoff, I am somewhat concerned about its performance under larger generation length budgets.

2. The entire motivation of the paper is built upon the observation that, for mathematical tasks, the initial prefix may directly determine the accuracy of the final answer. Regarding this, I have the following concerns: 1) This is an observation not rigorously proven theoretically; therefore, the theoretical justification for the effectiveness of PS-PPO, which is proposed based on this, is questionable. 2) This observation may not necessarily apply to alignment tasks. In the results table presented in the appendix, PS-PPO also underperforms compared to DAPO.

3. Theoretically, to compute $w^\theta_t(x)$ while also considering computational efficiency, cross timestep covariances are ignored and only the output head was taken into account, which may introduce significant bias.

---

> ### Author Rebuttal · Authors · 2026-03-31
>
> **[Weakness 1. Scalability to longer generations]**
>
> PS-PPO is designed to reduce gradient-update computation by backpropagating only through the sampled prefix, so its efficiency advantage becomes more pronounced as the maximum completion length grows. In additional experiments with larger maximum completion lengths, PS-PPO becomes substantially faster while remaining close in performance.
>
> Please see our response to Reviewer wrpc **(Weakness 1, Efficiency gains)**, where we report the full scaling table and breakdown.
>
> **[Weakness 2-1. Motivation for prefix]**
>
> This concern mainly targets **deterministic early truncation**, where later corrective tokens may be discarded. By using **stochastic cutoff** and reweighting **to preserve the full-sequence update** in expectation, PS-PPO does not discard later corrective tokens outright.
>
>  Please see our response to Reviewer wrpc **(Weakness 2 & Question 1, Motivation for prefix)**, where we address this point in detail.
>
> **[Weakness 2-2. Generality beyond math reasoning]**
>
> Appendix F was intended as a broader applicability check, since PS-PPO preserves the full-sequence update in expectation and is therefore **not tied to mathematical reasoning alone**. The method is designed to reduce gradient-update computation while remaining close in performance to full-update methods such as DAPO.
>
> The original appendix, however, under-emphasized the efficiency side by reporting reward without the corresponding time and memory analysis. We therefore now complement Appendix F with total training time and peak GPU memory, so that the results can be interpreted through the intended efficiency--performance trade-off rather than reward alone. Across both base models, PS-PPO substantially reduces training time and peak GPU memory while maintaining a favorable efficiency--performance trade-off.
>
> Please see our response to Reviewer wFj4 **(Weakness 2 & Question 3, Generality beyond math reasoning)**, where we report the full table.
>
> **[Weakness 3. Approximation bias in cutoff computation]**
>
> The key point is that these approximations affect only the cutoff computation, not the unbiasedness of the truncated estimator itself (Eq. 3). They are introduced to obtain a tractable timestep-wise surrogate for allocating limited update budget across timesteps.
>
>  Please see our response to Reviewer VA4E **(Weakness 2 & Question 3, Approximations in cutoff computation)** for details.
>
>
> **[Question 1. Why Eq. (3) leads to the variance objective]**
>
> Eq. (3) is not the variance-minimization objective; it is the unbiased truncated estimator. It replaces biased deterministic truncation with stochastic truncation plus inverse-probability reweighting.
>
> Once this correction is introduced, the remaining problem is how to choose the cutoff distribution under a fixed gradient-update budget B, since the reweighting also introduces additional variance. Sec. 3.3 addresses this by choosing  $\xi_{1:T}$ to control that added variance.
>
> Figure 1 provides the empirical motivation for truncation by suggesting substantial temporal redundancy in full-trajectory updates. The formulation then explains how to carry out this truncation without bias under a fixed gradient-update budget.
>
> **[Question 2. Explicit reasoning tokens and `<think>` spans]**
>
> To address the reviewer’s suggestion, we additionally evaluate DeepSeek-R1-Distill-Qwen-1.5B with a larger response-length budget of 8192, and report task performance, total training time, and simple statistics on how much of the explicit <think> span is retained by the sampled cutoff.
>
> **Table. Results on DeepSeek-R1-Distill-Qwen-1.5B with explicit `<think>` spans**
>
> | Method | Budget (B) | Math500 | AIME24 | AIME25 | Avg. | Total training time | Full `<think>` span retained (%) | Avg. retained fraction of `<think>` span (%) |
> |-|-:|-:|-:|-:|-:|-:|-:|-:|
> |DAPO| full |78.7|23.3|21.1|41.0|15.3 h|100|100|
> |PS-PPO|128|77.8|21.1|20.0|39.6|10.6 h|11.7|29.8|
> |PS-PPO|256|78.2|23.3|20.0|40.5|11.1 h|16.9|38.4|
> |PS-PPO|512|78.6|21.1|21.1|40.3|11.9 h|21.3|47.7|
>
> The results show that PS-PPO still provides a favorable efficiency--performance trade-off in this explicit-CoT setting. The `<think>`-retention analysis further suggests that strong performance does not require retaining the full `<think>` block in every sample, which is consistent with the intuition that not all later tokens need to receive gradient updates.

---

> > ### Author Rebuttal · Reviewer_1NwV · 2026-04-02
> >
> > Thank you for your responses. I believe my concerns remain, and I'll maintain my scores.

---

> > > ### Author Response · Authors · 2026-04-03
> > >
> > > We appreciate your time and consideration.
> > >
> > > In our rebuttal, we addressed the main concerns directly and concretely, including :
> > > - scalability and efficiency at longer generation lengths,
> > > - the motivation for using prefixes and the broader applicability of the method, and
> > > - the role of the approximations used in cutoff computation.
> > >
> > > We also supported our response with additional experiments and explanations. If there are **particular points** that you still view as unresolved, we would sincerely welcome any further details on those points, as that would help us better understand the remaining gap. We appreciate your engagement with the paper.

---

### Official Review · Reviewer_VA4E · 2026-03-12

**Soundness:** 2
**Presentation:** 3
**Significance:** 3
**Originality:** 3
**Overall Recommendation:** 4
**Confidence:** 4

**Summary:**

This paper proposes a novel critic-free policy optimization method called Prefix-Sampling Proximal Policy Optimization (PS-PPO). It uses the cutoff method to force the gradient on the prefix tokens to save the computation. This method introduces a unbiases stochastic truncation for prefix truncation and uses forward only proxy to approximate the score-norm to avoid the calculation of the whole model gradient. Empirical study shows the competitive performance but with a more efficient training process.

**Compliance With Llm Reviewing Policy:**

Affirmed.

**Final Justification:**

The rebuttal has addressed most of my concerns. I keep my positive score.

**Key Questions For Authors:**

- Can you explain the detailed settings such as the model and problem for Figure 1?

- Can you explain the compatibility of your assumption with the self-correcting capability? Can you provide more evidence showing the importance of the prefix token?

- Can you explain the approximation stage during the method derivation and provide some theoretical analysis for these approximations?

- Can you provide the explanation for the observation in Table 3 for the budget $B$?

**Limitations:**

yes

**Strengths And Weaknesses:**

**Strengths**


- This paper provides a novel approach called Prefix-Sampling Proximal Policy Optimization (PS-PPO) for finetuning the large language models with reinforcement learning. The proposed method relies on the observation that with the correct prefix, the model can generate the correct answers with high probability. PS-PPO aims to adopt the gradient update over the prefix token instead of the whole trajectory for more efficient training.

- The experiments on RLVR settings (math problem) demonstrate the competitive performance compared to the baselines and with lower training time in each step and peak memory. It shows the strong effectiveness of saving computation but keeping the performance in the math reasoning problems.

- The organization and presentation of this paper are clear and easy to follow.


**Weeknesses**

- The proposed method relies on the observation that with the correct prefix, the model can generate the correct answers with high probability. However, this assumption does not always hold in LLMs. The LLMs have the capability to self-correct when generating the chain of thoughts. For example, it can deny the previously generated tokens and generate the correct answers in the later tokens. The assumption of this paper does not seem to be appropriate. Besides, this observation only holds for math problems but may not be observed in instructions following or other tasks.

- The derivation of the proposed method involves a lot of approximation. In equation 4, the method directly ignores the cross-time step covariance. However, it can not easily neglect these covariances since the LLMs is genrating the tokens via autoregressive patterns based on the previous histories. Another approximation is the forward proxy. This approximation is only supported by the observation which is not sufficient.

- The experiments for text generation in Appendix F do not show the effectiveness of the proposed methods. It only demonstrates the inferior performance in the text generation task without time and memory analysis. It also needs to show that the prefix can determine the performance of the following trajectory in text generation tasks. The ablation study in Table 3 is not resonbale. It demonstrates that with a larger budget, the method will achieve poorer performance. It need more detailed explanation.

---

> ### Author Rebuttal · Authors · 2026-03-31
>
> **[Weakness 1 & Question 2. Motivation for prefix]**
>
> This concern mainly targets **deterministic early truncation**, where later self-correcting tokens may be discarded. PS-PPO instead uses **stochastic cutoff** and reweighting to maintain **unbiasedness**, so such later corrective effects are not discarded outright. Because the estimator preserves the full-sequence update in expectation, PS-PPO remains applicable across tasks, such as instruction following; **what may vary is not whether the method applies, but how much practical efficiency gain truncation provides**.
>
> Please see our response to Reviewer wrpc **(Weakness 2 & Question 1, Motivation for prefix)**, where we address this point in detail.
>
> **[Question 1. Figure 1 experimental setup]**
>
> Figure 1 is based on AIME 2024 and MATH-500 using Qwen2.5-Math-7B. For each problem, we first generate one full completion by greedy decoding. We then extract intermediate prefixes from that trajectory and, rollout 32 suffix continuations using the same model to estimate the prefix-conditioned success rate. The x-axis denotes prefix length normalized by the full completion length, and the y-axis reports the empirical success rate over the 32 sampled continuations. We will add these details explicitly to the figure caption and the experimental setup in the revision.
>
> **[Weakness 2 & Question 3. Approximations in cutoff computation]**
>
> A key clarification is that these approximations are used only to design the cutoff distribution $\xi_{1:T}$; they do not affect the unbiasedness of the truncated estimator itself.
>
> For the cross-time-step covariance term, the point is **not** that autoregressive dependencies disappear or that such covariances are exactly zero. Appendix B shows that the exact variance expansion is coupled across timesteps, so including the cross-time terms makes each timestep depend on the others and obscures the contribution of individual timesteps. Eq. (4) is therefore introduced as a tractable timestep-wise surrogate.
>
> For the forward-only proxy, the issue is similar. The exact weight
> $w_t^\theta(x)=E[\hat A_t^2\|\nabla_\theta \log \pi_\theta(o_t|s_t)\|^2|x]$
> depends on the full score norm, whose direct computation would require backward passes and would substantially offset the intended compute savings from truncation. What we need here, however, is not the exact score norm at every timestep, but a **tractable signal that preserves the relative importance of timesteps**. **Appendix C** therefore derives an output-head term analytically from forward-pass quantities, and **Appendix G** shows that this proxy is strongly correlated with the full-parameter score norm across timesteps. This is why we use the output-head proxy when computing $\xi_{1:T}$.
>
> Finally, Figure 3 supports the practical usefulness of these approximations. Under the same update budget, the optimized cutoff consistently outperforms the uniform, time-prior, and heuristic alternatives. This suggests that, although the cutoff computation uses approximations, they are informative enough to identify a more effective cutoff distribution in practice.
>
> **[Weakness 3. Generality beyond math reasoning]**
>
> Appendix F is intended as a broader applicability check, showing that PS-PPO is not restricted to binary rewards and can also be applied to continuous-reward text generation.
> Its purpose is not to claim that text generation must exhibit the same prefix-conditioned pattern as mathematical reasoning.
> Because the estimator remains unbiased, PS-PPO remains applicable beyond math, while the practical efficiency gain from truncation may vary across tasks.
>
> The original appendix, however, under-emphasized efficiency by reporting reward without the corresponding time and memory analysis. We therefore now add total training time and peak GPU memory for Appendix F.  Across both base models, PS-PPO substantially reduces training time and peak GPU memory, while maintaining a favorable efficiency--performance trade-off.
>
> Please see our response to Reviewer wFj4 **(Weakness 2 & Question 3, Generality beyond math reasoning)**, where we report the full table.
>
> **[Question 4. Interpreting the budget ablation]**
>
> Table 3 does not show monotonic degradation with larger budgets. Instead, performance improves from very small budgets and then saturates once the budget reaches a moderate range.
>
> When $B$ is too small, truncation is overly aggressive and reweighting increases variance, which can destabilize optimization. As $B$ increases, performance improves, but beyond a moderate range the added tokens are increasingly low-utility suffix tokens, so the gain saturates while update cost continues to grow.
>
> Thus, Table 3 reflects an efficiency--performance trade-off rather than a simple monotonic relationship. To make this more explicit, we additionally visualize the learning curves together with the truncation-induced variance across budgets (anonymous figure: https://anonymous.4open.science/r/PSPPO-DCCC).

---

> > ### Author Rebuttal · Reviewer_VA4E · 2026-04-03
> >
> > Thanks for the authors' detailed discussion and additional experiments, which have addressed most of my concerns. I hope the authors will incorporate these discussions into the revised manuscript. I keep my positive scores.

---

### Official Review · Reviewer_wrpc · 2026-03-12

**Soundness:** 3
**Presentation:** 3
**Significance:** 2
**Originality:** 2
**Overall Recommendation:** 4
**Confidence:** 3

**Summary:**

The paper proposes prefix-sampling PPO (PS-PPO), a technique which aims to make LLM RLHF more efficient (in terms of training time) by backpropagating policy gradients through truncated online rollouts.

They use a probabilistic truncation algorithm (such that the probability of truncation increases monotonically along any rollout) and provide a theoretically principled algorithm to compute these probabilities.

The paper demonstrates gains in training time over existing token-selective RLHF methods (such as S-GRPO), while maintaining competitive performance on math reasoning benchmarks.

**Compliance With Llm Reviewing Policy:**

Affirmed.

**Final Justification:**

Rebuttal addressed both concerns hence I am increasing the score.

**Key Questions For Authors:**

My main concern is regarding the central claim (see for eg. lines 60-63 on the left side in page 2) that "policy gradient updates applied to later tokens contribute little additional learning signal and are largely reduntant".

 I believe this might be false in rollouts where the LLM self-corrects, in which case we certainly want to reinforce the later tokens more.

Could the authors please help clarify this?

**Limitations:**

yes

**Strengths And Weaknesses:**

Strength.

1. The paper is well written and easy to follow.
2. The theory behind computing the truncation probabilities seems insightful and could have applications beyond RLHF.

Weaknesses.
1. The gains in training time seem modest over existing token-selective methods (1.77 for PS-PPO vs 2.66 for S-GRPO) and the performance is at best competitive with the baselines, but the PS-PPO algorithm seems fairly complicated (both conceptually and implementation wise), Such algorithms might be less likely to stand the test of time

2. The paper claims that later tokens are strictly less likely to contribute to the advantage than earlier tokens, but I tend to disagree with this. For example, it is known that LLMs trained with RL can develop self-correction abilities (where it corrects its own earlier mistakes later in the COT), in which case reinforcing later tokens should certainly be useful.

---

> ### Author Rebuttal · Authors · 2026-03-31
>
> **[Weakness 1. Efficiency gains]**
>
> Our goal is **not to maximize accuracy** in isolation, but to improve the **efficiency-performance trade-off** of critic-free RLHF by reducing the cost of the gradient-update stage while preserving competitive performance.
>
> Under this objective, the gains are already significant at the setting reported in the paper. At $T_{\max}=1024$, PS-PPO reduces training time per step from 2.66s to 1.77s versus S-GRPO (about **33% faster**) and from 3.23s to 1.77s versus DAPO (about **45% faster**), while remaining competitive with strong critic-free baselines in pass@1 accuracy.
>
> Moreover, the benefit grows with completion length, which is precisely the regime PS-PPO is designed for. In our additional experiments, at $T_{\max}=2048$, PS-PPO is about **2.0x** faster than S-GRPO and **2.4x** faster than DAPO; at $T_{\max}=4096$, the gap increases to **2.8x** and **3.3x**, respectively. Across these longer-completion settings, PS-PPO remains competitive with DAPO in performance.
>
> Regarding complexity, PS-PPO is more structured than simple masking, but it remains lightweight compared with alternatives that require extra learned components. It introduces **no critic, no auxiliary model, and no extra rollouts**. The additional machinery is limited to a prompt-wise cutoff design together with truncated prefix backpropagation and reweighting, which is what enables compute reduction without the bias of naive truncation.
>
> **Scaling with maximum completion length.**
> We compare PS-PPO with S-GRPO and DAPO while varying the maximum completion length, and report training time per step together with accuracy.
>
> | $T_{\max}$ | Method | Training time / step | Math500 | AIME24 | AIME25 | Avg. |
> |---|---|---:|---:|---:|---:|---:|
> | 1024 | **PS-PPO** | 1.77 ± 0.02 | 85.0 | 16.7 | 13.3 | 38.3 |
> | 1024 | S-GRPO | 2.66 ± 0.01 | 84.7 | 16.7 | 10.0 | 37.1 |
> | 1024 | DAPO | 3.23 ± 0.01 | 85.2 | 16.7 | 10.0 | 37.3 |
> | 2048 | **PS-PPO** | 2.02 ± 0.03 | 86.2 | 23.3 | 16.7 | 42.1 |
> | 2048 | S-GRPO | 4.03 ± 0.03 | 85.6 | 20.0 | 13.3 | 39.6 |
> | 2048 | DAPO | 4.93 ± 0.03 | 86.4 | 23.3 | 16.7 | 42.1 |
> | 4096 | **PS-PPO** | 2.39 ± 0.04 | 86.4 | 23.3 | 16.7 | 42.1 |
> | 4096 | S-GRPO | 6.70 ± 0.04 | 85.8 | 20.0 | 13.3 | 39.7 |
> | 4096 | DAPO | 7.78 ± 0.04 | 86.6 | 23.3 | 16.7 | 42.2 |
>
>
> **Training-time breakdown across maximum completion lengths.**
> We report the time spent on computing $\xi_{1:T}$, forward, backward, and other overheads at $T_{\max} \in \{1024, 2048, 4096\}$, excluding rollout/generation.
>
> | $T_{\max}$ | Method | Computing $\xi_{1:T}$ | Forward | Backward | Other | Training time / step |
> |---|---|---:|---:|---:|---:|---:|
> | 1024 | **PS-PPO** | 0.43 ± 0.02 | 0.32 ± 0.04 | 1.01 ± 0.14 | 0.01 ± 0.01 | 1.77 ± 0.02 |
> | 1024 | S-GRPO | N/A | 0.82 ± 0.07 | 1.81 ± 0.10 | 0.03 ± 0.02 | 2.66 ± 0.01 |
> | 1024 | DAPO | N/A | 1.11 ± 0.04 | 2.10 ± 0.13 | 0.02 ± 0.01 | 3.23 ± 0.01 |
> | 2048 | **PS-PPO** | 0.50 ± 0.03 | 0.39 ± 0.05 | 1.13 ± 0.12 | 0.01 ± 0.01 | 2.02 ± 0.03 |
> | 2048 | S-GRPO | N/A | 1.31 ± 0.08 | 2.68 ± 0.14 | 0.04 ± 0.02 | 4.03 ± 0.03 |
> | 2048 | DAPO | N/A | 1.78 ± 0.06 | 3.12 ± 0.15 | 0.03 ± 0.01 | 4.93 ± 0.03 |
> | 4096 | **PS-PPO** | 0.61 ± 0.04 | 0.48 ± 0.05 | 1.30 ± 0.13 | 0.02 ± 0.01 | 2.39 ± 0.04 |
> | 4096 | S-GRPO | N/A | 2.18 ± 0.10 | 4.47 ± 0.18 | 0.05 ± 0.02 | 6.70 ± 0.04 |
> | 4096 | DAPO | N/A | 2.56 ± 0.09 | 5.18 ± 0.20 | 0.04 ± 0.01 | 7.78 ± 0.04 |
>
> **[Weakness 2 & Question 1. Motivation for prefix]**
>
> Later tokens can indeed carry useful learning signal, especially when the model self-corrects during generation. A **deterministic early truncation** can miss such later corrective effects, because the update no longer reflects tokens that appear after the truncation point. PS-PPO is designed to avoid that issue. It uses **stochastic cutoff** sampling and reweighting, so the truncated update remains **unbiased** in expectation with respect to the full-sequence update, rather than deterministically discarding later corrective tokens.
>
> This is also consistent with our empirical results. Under the same update budget, fixed-length truncation performs worse, while PS-PPO with an optimized cutoff performs better than simpler truncation strategies.

---

> > ### Author Rebuttal · Reviewer_wrpc · 2026-04-03
> >
> > Thanks for the additional experiments and discussion about truncation length. I have increased the score to 4.

---

### Decision · Program_Chairs · 2026-04-30

**Decision:**

Accept (regular)

**Comment:**

This paper proposes PS-PPO, a compute-efficient critic-free RLHF method that samples a stochastic cutoff prefix for each trajectory and applies an unbiased importance-weighted gradient update. The approach reduces training time and peak GPU memory while maintaining competitive performance on math reasoning benchmarks.

Three reviewers (wrpc, VA4E, wFj4) initially raised concerns: (1) the assumption that early prefixes determine final rewards may ignore late-sequence self-corrections; (2) approximations in cutoff computation could introduce bias; (3) experiments are limited to math reasoning; (4) novelty relative to token-selective methods (e.g., S-GRPO) is incremental. All three gave “weak accept” scores after rebuttal.
The authors provided a strong rebuttal with additional experiments: scaling to 4096 tokens (showing 2.8–3.3× speedups), results on DeepSeek-R1-Distill with explicit <think> spans, text-generation tasks with time/memory analysis.

The paper presents a technically sound, empirically grounded efficiency improvement for critic-free RLHF. The unbiased truncation framework and budget-aware cutoff optimization are principled contributions. While the method relies on an empirical observation about prefix informativeness (which may not hold universally) and involves approximations, the rebuttal convincingly demonstrates its practical gains across longer sequences and a second domain. The minority dissent does not outweigh the overall positive reception from three reviewers. Given the solid empirical results and clear efficiency benefits, I recommend weak accept.